

# Path for recovery: an ecological overview of the Jambato Harlequin Toad (Bufonidae: *Atelopus ignescens*) in its last known locality, Angamarca Valley, Ecuador

Mateo A. Vega-Yánez[1,2], Amanda B. Quezada-Riera[3], Blanca Rios-Touma[4], María del Carmen Vizcaíno-Barba[3], William Millingalli[3], Orlando Ganzino[3], Luis A. Coloma[3,5], Elicio E. Tapia[6], Nadine Dupérré[6], Mónica Páez-Vacas[3,5,7], David Parra-Puente[3,8], Daniela Franco-Mena[2], Gabriela Gavilanes[2], David Salazar-Valenzuela[3,7,9], Carlos A. Valle[10] and Juan M. Guayasamin[1,2,3]

[1] Universidad San Francisco de Quito USFQ, Colegio de Ciencias Biológicas y Ambientales COCIBA, Maestría en Ecología Tropical y Conservación, Quito, Ecuador

[2] Universidad San Francisco de Quito USFQ, Colegio de Ciencias Biológicas y Ambientales COCIBA, Laboratorio de Biología Evolutiva, Calle Diego de Robles s/n y Pampite, Campus Cumbayá, Quito, Ecuador

[3] Alianza Jambato, Las Casas, Quito, Ecuador

[4] Grupo de Investigación en Biodiversidad, Medio Ambiente y Salud (BIOMAS). Facultad de Ingenierías y Ciencias Aplicadas, Universidad de Las Américas, Vía Nayón S/N, Campus UDLAPARK, Quito, Ecuador

[5] Centro Jambatu de Investigación y Conservación de Anfibios, Fundación Jambatu, San Rafael, Quito, Ecuador

[6] Leibniz-Institute for the Analysis of Biodiversity Change (LIB), Museum of Nature, Hamburg, Germany

[7] Centro de Investigación de la Biodiversidad y Cambio Climático (BioCamb), Universidad Indoamérica, Machala y Sabanilla, Quito, Ecuador

[8] Fundación de Conservación Jocotoco, Quito, Ecuador

[9] Ingeniería en Biodiversidad y Recursos Genéticos, Facultad de Ciencias de Medio Ambiente, Universidad Indoamérica, Machala y Sabanilla, Quito, Ecuador

[10] Universidad San Francisco de Quito USFQ, Colegio de Ciencias Biológicas y Ambientales COCIBA, Quito, Ecuador

Corresponding authors
Mateo A. Vega-Yánez,
mvegayanez183@gmail.com
Juan M. Guayasamin,
jmguayasamin@gmail.com

## ABSTRACT

The Jambato Harlequin toad (*Atelopus ignescens*), a formerly abundant species in the Andes of Ecuador, faced a dramatic population decline in the 1980s, with its last recorded sighting in 1988. The species was considered Extinct by the IUCN until 2016, when a fortuitous discovery of one Jambato by a local boy reignited hope. In this study, we present findings from an investigation conducted in the Angamarca parish, focusing on distribution, abundance, habitat preferences, ecology, disease susceptibility, and dietary habits of the species. In one year we identified 71 individuals at different stages of development in various habitats, with a significant presence in agricultural mosaic areas and locations near water sources used for crop irrigation, demonstrating the persistence of the species in a complex landscape, with considerable human intervention. The dietary analysis based on fecal samples indicated a diverse prey selection, primarily comprising arthropods such as Acari, Coleoptera, and ants. Amphibian declines have been associated with diseases and climate change; notably, our study confirmed the presence of the pathogen *Batrachochytrium dendrobatidis* (*Bd*), but, surprisingly, none of the infected Jambatos displayed visible signs of illness. When analyzing climatic patterns, we found that there are climatic differences between historical localities and

Angamarca; the temporal analysis also exposes a generalized warming trend. Finally, in collaboration with the local community, we developed a series of management recommendations for terrestrial and aquatic environments occupied by the Jambato.

# INTRODUCTION

*Atelopus ignescens* (*Cornalia, 1849*), commonly called Jambato Harlequin toad (Jambato, hereafter), is an endemic and historically abundant species of the Andes of Ecuador. Its common name derives from the Kichwa word ''jampatu'', ''hampatu'' or ''hambatu'' (depending on the spelling system), which means frog/toad, because it was practically the image of an adult anuran for humans inhabiting these areas. Its historical distribution included the Inter-Andean valleys and Páramos of central-northern Ecuador, from the province of Imbabura to the provinces of Chimborazo and Bolívar at elevations of 2,800 to 4,200 m a.s.l. (*Coloma, Lötters & Salas, 2000*; *Ron et al., 2003*; *Guayasamin et al., 2010*). Some records of abundance and autecology date back to 1864 when the Spaniard naturalist Marcos Jiménez de la Espada observed ''thousands of individuals in the grassy and humid meadows near streams, ponds and lakes'' in the Páramos of Antisana volcano (*Jiménez de la Espada, 1875*:146). In addition, he describes their reproductive habits such as that Jambatos perform an axillary amplexus and that the palms of their feet and hands become thicker and more extended, possibly preparing for egg laying in the water (*Jiménez de la Espada, 1875*). Years later, in 1981, at the same site Juan Black counted up to 50 individuals per square meter (*Coloma & Quiguango-Ubillús, 2018*). Similarly, in 1981, on the shores of the Limpiopungo lagoon, Carlos Valle recorded between 12 and 60 individuals in an area of 420 square meters. He also reported that the activity of the Jambato was related to temperature and observed a diet based mainly on Diptera and Coleoptera (*Valle, 1984*). There are also reports of massive migrations in the provinces of Tungurahua and Bolívar, where numerous individuals were crushed on the road (*Peters, 1973*). In May 1985, according to Luis Coloma's field notes, a massive migration was recorded in the Páramo of Chimborazo volcano that left ''thousands of toads crushed along 1 km of the Guaranda-Ambato road''.

The Jambato was not restricted to relatively pristine and protected environments (*e.g.*, Antisana and Cotopaxi volcanoes), but the species was also common in lands modified by humans. For example, the Jambato was frequent in grasslands, pastures, and some parks of Andean cities such as Quito, Latacunga, and Ambato (*Ron et al., 2003*). The name of the latter city originates from the common name of the toad and appears as ''Hambato'' in maps and official documents (*Maldonado, 1750*). Marcos Jiménez de la Espada even referred to this place as the ''Hill of Hambato'' (*Jiménez de la Espada, 1875*). Today, there is a public monument in honor of this species in Ambato, a symbol of its cultural importance.

*Atelopus ignescens* suffered dramatic population crashes in the 1980s; its last record before the 2016 rediscovery, was on March 30th, 1988, in Oyacachi (*Coloma, Lötters & Salas, 2000*;

*Ron et al., 2003*). Before that date, the species was present in 64% of the localities recorded throughout its distribution (*Ron et al., 2003*). The contrast of presence-absence data, combined with extensive recent field surveys, suggested that the species was extinct (*Ron et al., 2003*; *Bustamante, Ron & Coloma, 2005*). During a similar timeframe, numerous species of *Atelopus* also disappeared from the Andes (*La Marca et al., 2005*; *Lötters et al., 2023*). The possible causes linked to the population declines and disappearance of *Atelopus* point mostly to two non-exclusive hypothetical causes: unusual changes in climate and the spread of amphibian skin fungus, caused by *Batrachochytrium dendrobatidis* (*Berger et al., 1998*; *La Marca et al., 2005*; *Rohr et al., 2008*; *McCaffery, Richards-Zawacki & Lips, 2015*; *Gómez-Hoyos et al., 2018*; *Scheele et al., 2020*). In the case of *A. ignescens*, the evidence shows that a year before its disappearance the climate was unusually warm and dry (*Ron et al., 2003*). Also, those crashes coincide with the first reports of chytridiomycosis in Ecuador, diagnosed in *A. bomolochos* in 1980 (*Ron & Merino, 2000*). However, in analyses of 89 museum specimens of *A. ignescens*, none showed the pathogenic fungus (*Merino-Viteri, 2001*; *Ron et al., 2003*). Although all evidence suggested that the Jambato was extinct, searches for the species continued, led by one of us (Coloma) and the indefatigable efforts of Giovanni Onore for almost three decades. In 2016, in the community of Angamarca, Cotopaxi province, a local boy named David Jailaca, found a Jambato in an alfalfa field, and communicated the news to Onore and Coloma. Since then, the Jambato has become an emblematic species that exemplifies the amphibian crisis in Ecuador, but also the hope for recovery. This event led Centro Jambatu (http://www.anfibiosecuador.ec/) to conduct an ex-situ breeding program as an emergency measure to save the species from extinction (*Coloma, 2016*).

It took five more years to start an in-situ conservation program, which includes a research component to understand the ecology and local dynamics of the population of *A. ignescens* in the field. Here, we present the results of this study, focusing on the following main questions: (i) What are the current occupation areas of the Jambato within the Angamarca River basin? (ii) Which are the characteristics of the terrestrial and aquatic habitats used by the Jambato? (iii) Is the Jambato reproducing in its natural environment? (iv) Is the chytrid fungus infecting the Jambato? (v) Which management recommendations are drawn from the obtained ecological data?

## MATERIALS & METHODS

### Ethical statement

The research was conducted under permits issued by the Ministerio de Ambiente, Agua y Transición Ecológica No. MAATE-DBI-CM-2022-0245 where ethical approvals can be found. In addition, the research adhered to the guidelines outlined for the use of live amphibians and reptiles in field research, as specified in the guidelines proposed by *Beaupre et al. (2004)*.

### Study area

The 2016 rediscovery of *Atelopus ignescens* took place in the community of San Pablo, Angamarca parish in the Cotopaxi province, Ecuador, at 2,812 m a.s.l. This site is located
in the western foothills of the Andes Mountains. Geologically, the area is part of the Macuchi formation, which is considered a submarine volcanic arc, formed by subacute volcaniclastic rocks with some lavas and subvolcanic intrusions, dating from the Early to Middle Eocene (*Hughes & Bermúdez, 1997*). The Macuchi unit is conformably overlain by rocks of the Angamarca group which is a siliciclastic basin-fill sequence of Paleocene to Eocene age (*Vallejo et al., 2009*). The study area has a landscape with an elevational range from 2,000 to 4,064 m a.s.l. The parish is crossed longitudinally by the Angamarca River to the west and by the Guambaine River to the east. The Angamarca hydrological system belongs to the Guayas River basin and its climate is divided into a rainy season from October to May and a dry season from June to mid-September (Fig. S1). The average annual temperature ranges between 4.7 to 14.3 °C, while annual precipitation varies between 613–1,594 mm (*Fick & Hijmans, 2017*) (Fig. 1). A limitation of the WorldClim climatic data is that they are based on interpolations and, given the topography of the Andes, some of these interpolations might have errors.

## Fieldwork and data collection

The field surveys were conducted using a free-search method, which consists of collecting information without time or location constraints (*Angulo et al., 2006*). This approach is opportunistic because individuals are spontaneously collected. This methodology allowed us to obtain information on the absence/presence of the species on a relatively large geographic area, as well as exploring potentially new occupation areas of the species. We conducted 12 field trips of six days each, from December 2021 to December 2022. On average, each field trip consisted of a team of two people conducting searches of eight hours per day. Most searches were performed during the day because *Atelopus ignescens* is a diurnal species (*Coloma & Quiguango-Ubillús, 2018*); however, to only search for resting adult individuals, we carried out five nocturnal trips between 19h00 and 22h00.

Based on the topographic and hydrological characteristics of the Angamarca parish, we selected several searching areas, mainly those that have permanent streams, rivulets, or running water bodies, covering the largest possible surface, and surveying all available habitat types, from native forest and brush patches to crops and pastures, as well as rivers and streams for tadpoles. We also surveyed the accessible elevational range of the area, from 2,000 to 4,064 m a.s.l. Since our sampling was conducted over one year, we were able to obtain information on the species in both the rainy and dry seasons (Table S1).

For each observed Jambato, we recorded the following information: date and time of capture, locality (latitude, longitude, and elevation), habitat and microhabitat, snout-vent length (SVL) for adults and juveniles, body length (BL) and tail length (TAL) for tadpoles, activity of the individual at the time of observation, and weather conditions. We also conducted skin swabbing to test for the presence of the fungus *Batrachochytrium dendrobatidis* (*Bd*). Finally, we photographed each individual for photo-identification purposes. Extreme care was taken when handling individuals; we used different nitrile gloves for each individual, and this procedure was repeated in the same location when different Jambatos were found.

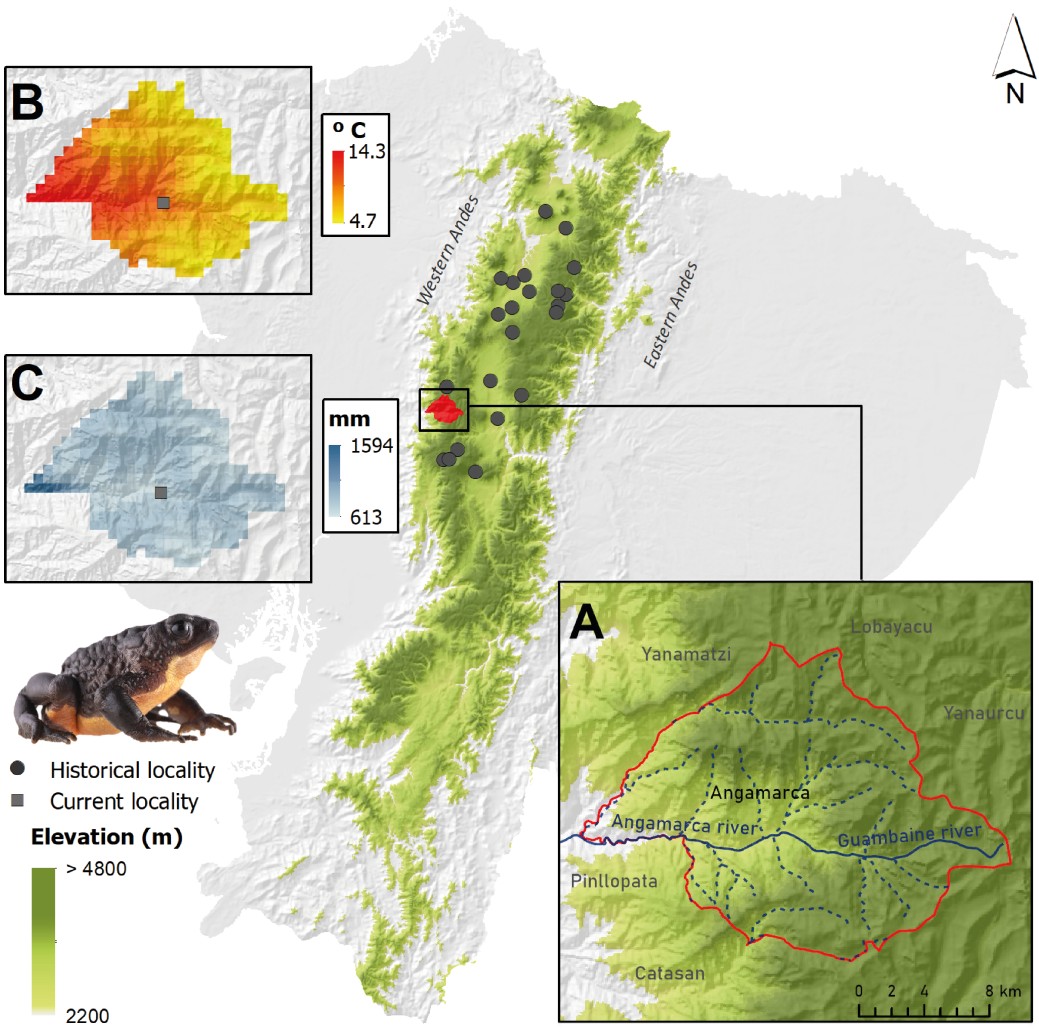

**Figure 1** **Distribution of the Jambato, *Atelopus ignescens*, in Ecuador, showing historical records (gray circles) and its current occurrence in the Angamarca Valley (square).** (A) Location of the Angamarca parish and river basin. (B) Average annual temperature in Angamarca parish. (C) Mean annual precipitation in Angamarca parish. Images prepared by Mateo A. Vega-Yánez.

## Distribution and landscape use of the Jambato

To assess the relationship of *Atelopus ignescens* with the landscape, we downloaded the landuse coverage map at a 1: 25,000 scale from the geoportal SIGTIERRAS (http://www.sigtierras.gob.ec/). This layer classifies landscape into six landuse classes (populated area, agricultural mosaic, shrub vegetation, Páramo, grassland, and herbaceous vegetation). We used the IUCN Species Mapping Tools v2021 to measure the Extent of Occurrence (EOO) of *Atelopus ignescens*, using a minimum convex polygon (*IUCN Standards Petitions Committee, 2022*). Additionally, we created a heat map in ArcGis Pro to determine the record density during the sampled period (Dec 2021 to Dec 2022). Likewise, using the R package *landscapemetrics* (*Hesselbarth et al., 2019*), we calculated one metric of composition and seven metrics of landscape configuration (Table 1). For this analysis, we

**Table 1  Description of the landscape metrics calculated for *Atelopus ignescens* in the study area.**

| Code | Name of metric | Description | Unit |
|------|----------------|-------------|------|
| PLAND | Percentage of landscape | It is an area and edge metric. It is the percentage of the landscape belonging to class i | % |
| CA | Class area | It is an area and edge metric that quantifies the total area of the class | ha |
| NP | Number of patches | It is an aggregation metric that quantifies the number of patches that have a class | None |
| PD | Patch density | An aggregation metric that quantifies patch density | Number per 100 Ha |
| LPI | Largest patch index | It is an area and edge metric that quantifies the index of the largest patch | % |
| DIV | Division index | It is an aggregation metric that quantifies the index of division that has class present in the landscape | Proportion |
| AI | Aggregation index | It is an aggregation metric that is based on adjacency for the class | % |
| ENN MN | Mean of Euclidean nearest-neighbor distance | It is an aggregation metric that quantifies the mean Euclidean distance from the nearest neighbor | m |

excluded the ''populated area'' class, which is based on buildings and, therefore, does not represent a feasible habitat for the species.

## Characterization of aquatic environments

To assess if environmental variables from streams had a role in tadpole presence and abundance, we chose 10 sites (Table S2), eight with tadpole and adult presence in their surrounding areas and two where no Jambatos were recorded during the monitoring year. At each site, we measured the following water parameters: pH, conductivity, oxygen concentration, total suspended solids, ammonium, nitrates, and phosphates, using standard methods (APHA, AWWA, WEF 2012). We used petrifilm to assess coliform bacteria presence and concentration (3MTM). Additionally, we collected a sample of benthic substrate to assess chlorophyll in the laboratory as a proxy of benthic algae abundance. We characterized each site by pebble counts (*Kondolf, 1997*), and the application of the Fluvial Habitat Quality index (IHF) and the Quality of the Andean Riparian Area (QBR) (*Acosta et al., 2009*). Additionally, we obtained one multi-habitat sample per stream, using a D-net, sampling all the possible microhabitats during three minutes. The sample was preserved in ethanol 90% and analyzed in the laboratory for macroinvertebrate identification. With these data, we calculated the Andean Biotic Index (ABI) (*Ríos-Touma, Acosta & Prat, 2014*) to assess the biological integrity of the sites.

## Photo-identification for recapture/photographic records

Using photo-identification, we tested the feasibility of identifying individual Jambatos through external phenotypic characteristics such as markings, design patterns, and

coloration (*Elgue et al., 2014*; *Gardiner et al., 2014*; *Gould et al., 2021*). Photo-identification has the straight-forward benefit of individual identification, avoiding invasive techniques (*e.g.*, toe-clipping; *Davis & Ovaska, 2001*), and is especially important when working with threatened species (*Gould et al., 2021*). Each Jambato was photographed, dorsally and ventrally, against a white background, with a Canon 6D Mark II (26.2 MP) camera, with a Sigma 105 mm lens. The photos were taken at the location where Jambatos were found, and the individuals were released after being photographed. The photos were analyzed by Amanda B. Quezada-Riera in Adobe Photoshop, comparing each individual with another after in search for individual features. All photographs are stored at Laboratorio de Biología Evolutiva (LBE) of the Universidad San Francisco de Quito (USFQ) and are available at request; additionally, photographs are published on iNaturalist (https://ecuador.inaturalist.org/projects/atelopus-ignescens-angamarca).

## Incidence and prevalence of *Batrachochytrium dendrobatidis* (*Bd*)

Jambatos captured in the field were swabbed a total of 30 times (five times on the back, five on the belly, five on each foot, and five on each hand), following the procedures described in *Hyatt et al. (2007)*. Swabs were stored in sterile Eppendorf tubes of 1.5 ml with ethanol at 96%; each individual was handled with a new set of gloves to avoid transmission of the fungus between individuals and localities (*Brem & Lips, 2008*). We processed 41 swabs at LBE-USFQ. DNA was extracted using the protocol designed by *Peñafiel et al. (2019)*. The presence/absence of *Bd* was assessed by using two specific primers developed by *Annis et al. (2004)*; the 19-nucleotide primer Bd1a (CAGTGTGCCATATGTCACG), which lies in a conserved 97 bp region of the ITS1, and the 20 nucleotides primer Bd2a (CATGGTTCATATCTGTCCAG) in a conserved 50 bp region of the ITS2 of *B. dendrobatidis*. In infected individuals, single bands of 300 bp were visualized on 1% agarose gel (*Annis et al., 2004*). We performed an initial PCR in 25 µl reactions using 0.5 µl deoxynucleotide triphosphate (dNTP) (10 mM), 2.5 µl PCR buffer 5X (200 mM Tris-HCl (pH 8.4), 2 µl MgCl2 (50 mM), 1µl of each primer (10 µM), 0.2 µl of Taq DNA polymerase, 1 µl template DNA (100 ng/ul) and 16.25 µl of H2O. Polymerase Chain Reaction (PCR) amplification protocols are presented in Table 2. Once we obtained these amplicons, we did a second PCR using them as DNA templates and changing the amount of MgCl2 to 1.75 µl and following the thermocycling conditions, changing to 20x the number of cycles.

## Diet of *Atelopus ignescens*

The diet for the species was based on analysis of fecal samples of 27 adult individuals found on May 10, 2016, alongside the Angamarca River and its tributaries. Jambatos were located at elevations between 2,500 to 2,861 m a.s.l. They were placed in a plastic container and brought to the lab. The feces were recollected in 70% ethanol and sorted under Leica M125 C dissection microscope. Animal body parts (*e.g.*, abdomen, elytra, carapace, chela, or body segments) or complete specimens (*e.g.*, acari or larvae) within the feces were identified to the lowest taxonomic level possible (Table S3). For Acari and larvae we measured total body length, while in Coleoptera only the elytra total length was measured; for other groups we measured various body parts (*e.g.*, abdomen, carapace, heads) (Table S4). All

**Table 2** Thermocycling conditions used to identified *Batrachochytrium dendrobatidis* presence or absence using polymerase chain reaction (initial - PCR).

| STEP | Temperature | Time | Cycles |
|---|---|---|---|
| Initial denaturation | 93 °C | 10 min | X 1 |
| Denaturation | 93 °C | 45 s | |
| Annealing | 60 °C | 45 s | X 30 |
| Extension | 72 °C | 1 min | |
| Final extension | 72 °C | 10 min | X 1 |
| | 4 °C | ¥ | |

measurements were done using a Leica M205A microscope with Leica Application Suite X (measurements are presented in mm). The abbreviation ECFN is an identifier number used to track the samples. Morphospecies were imaged with a custom-made BK Plus lab System by Dun Inc. Palmyra, PA, USA with an integrated Canon camera, a macro lens (65 mm) with the stacking software Zerene version 1.04 (Zeren Systems LLC, WA, USA). SEM images were taken using a Hitachi tabletop Microscope TM4000 plus.

## Climate analysis of the historical localities and Angamarca

We analyzed the temperature and precipitation of 23 localities (22 with historical and one with present-day records of the Jambato). We obtained historical monthly weather data from the WorldClim version 2.1 for 1960–2021 (*Fick & Hijmans, 2017*; *Harris et al., 2020*) with a spatial resolution of 2.5 min, which represents an area of ~4.6 km$^2$. We calculated the mean temperature and precipitation values for each month (732 months) for each locality. To analyze the variation in temperature and precipitation, we performed a Wilcoxon Rank-Sum Test for multiple comparisons between Angamarca and historical localities. To understand the variations of temperature and precipitation over time, we defined three time periods based on the documented population dynamics of the species (*Ron & Merino, 2000*; L. A. Coloma, field notes, 1985): (1) Pre-decline (1960–1980), (2) Population crashes (1981–1990) and (3) Current (1991–2021). For each period we calculated the mean and standard deviation of temperature and precipitation in R software. Using the *ggplot2* package we created boxplots to show the variation of the data for each variable (*R development core team, 2022*).

## RESULTS

### Distribution and abundance patterns

During one year searching for *Atelopus ignescens* at different elevations and habitat types within the Angamarca parish, we found 71 individuals in different stages of development (adults, sub-adults, juveniles, tadpoles; 63 adults/subadults (32 males, 31 females), two juveniles, and six tadpoles (Fig. 2). The identification at the individual level was possible through photo-identification of ventral coloration patterns (Fig. 3). Photo-identification was not performed on tadpoles, but it was considered that they were probably different individuals based on the location where they were found. The highest number of records was in December 2021, with 23 individuals; we did not find any Jambato during the driest

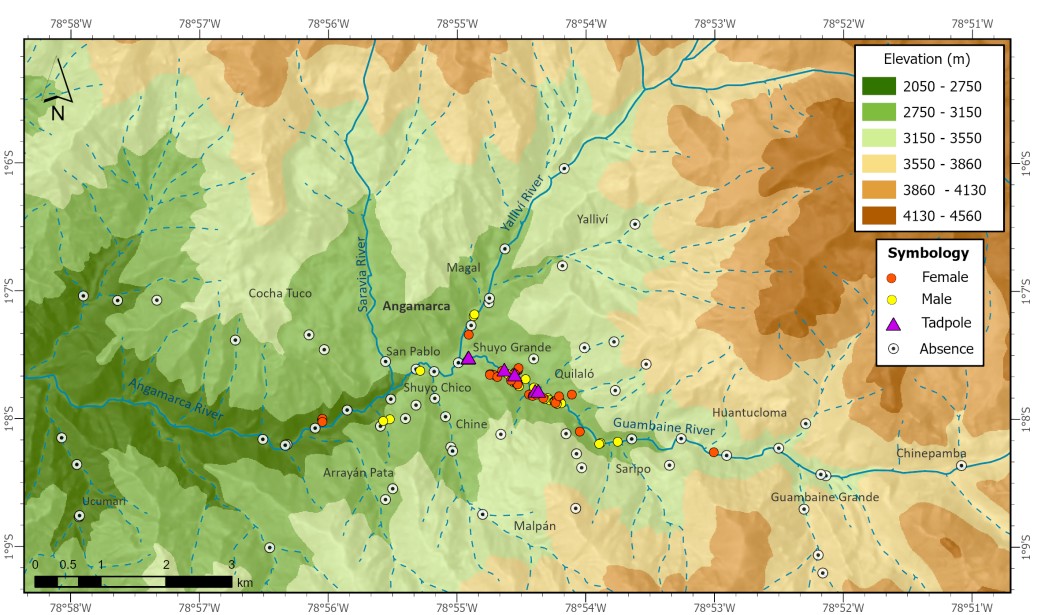

**Figure 2** Records of presence and absence of the Jambato toad, *Atelopus ignescens,* documented during field work in the Angamarca River basin. Image prepared by Mateo A. Vega-Yánez.

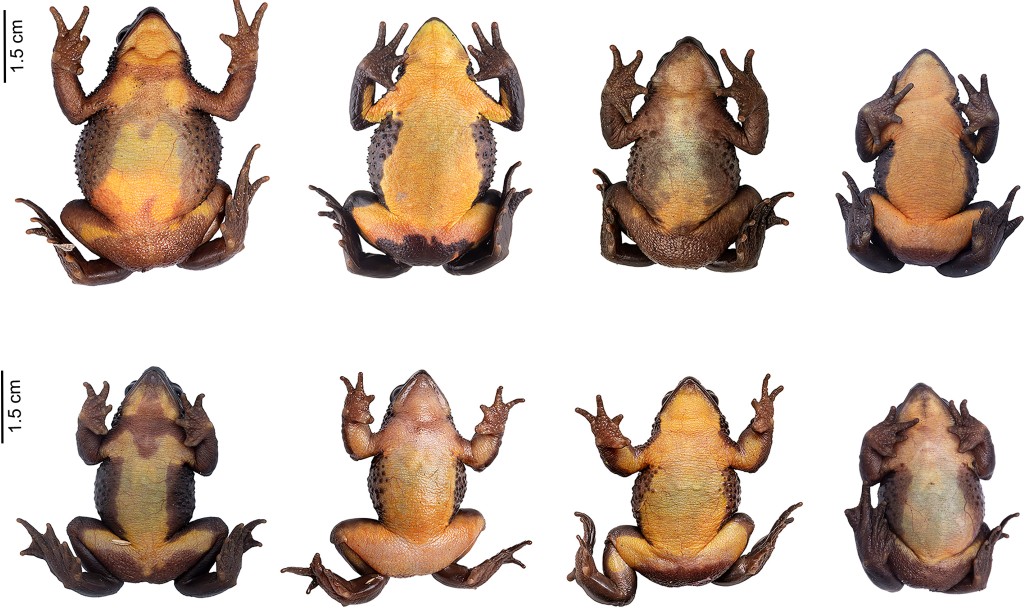

**Figure 3** Ventral variation in *Atelopus ignescens.* Top row: females. Bottom row: males. Photos by Amanda B. Quezada-Riera.

months (August and September 2022). The land use classes where we found *A. ignescens* are agricultural mosaic, shrub vegetation, Páramo, grassland, and herbaceous vegetation. All tadpoles were found in different parts of the Guambaine River (Fig. 4).

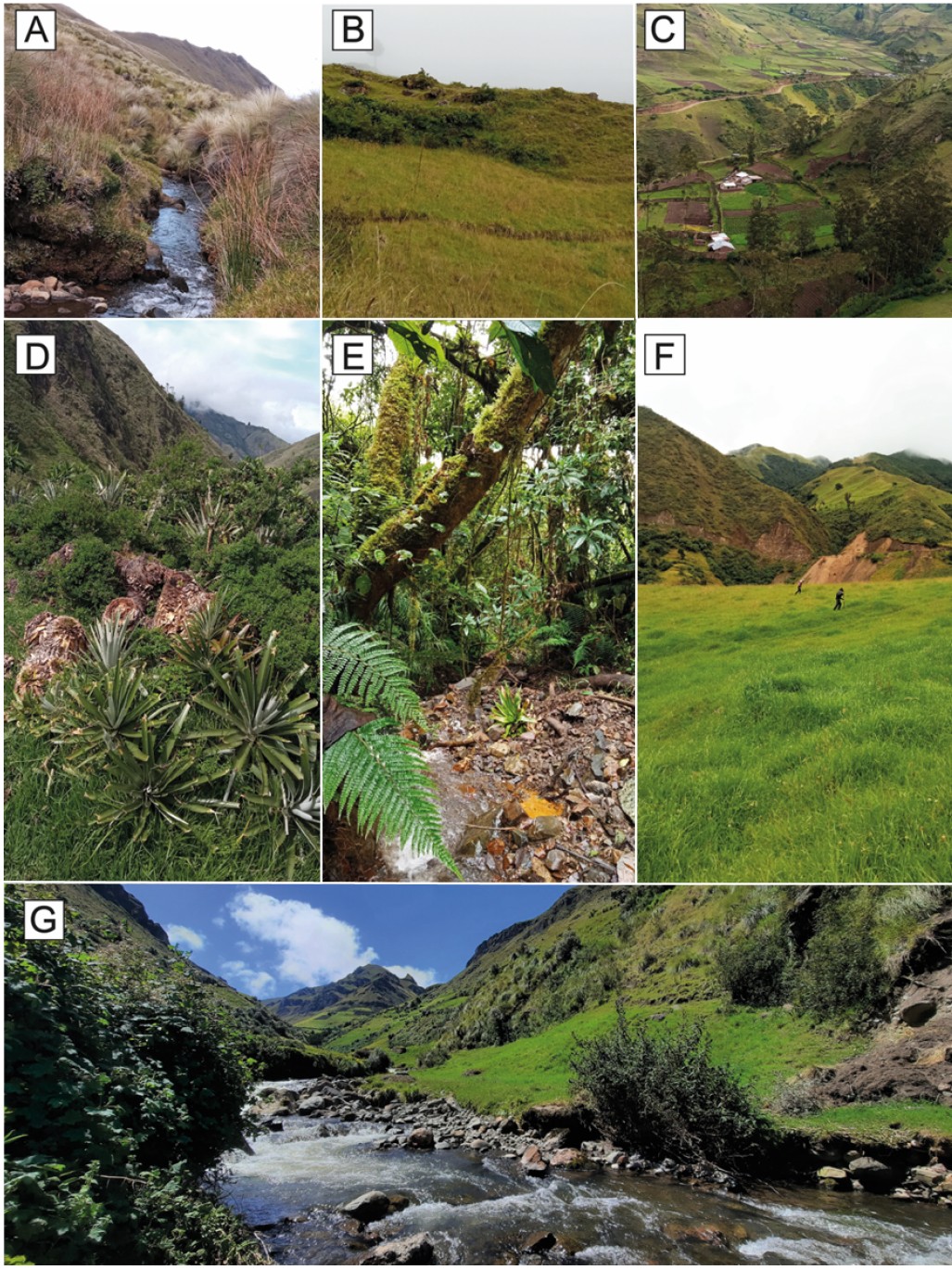

**Figure 4  Land use classes occupied by the Jambato at the Angamarca River basin.** (A) Páramo. (B) Shrub and herbaceous vegetation. (C) Agricultural mosaic. (D) Shrub vegetation. (E) Montane Forest. (F) Grassland for cattle farming. (G) Guambaine River. Photographs A–F prepared by Mateo A. Vega-Yánez, and G by Amanda B. Quezada-Riera.

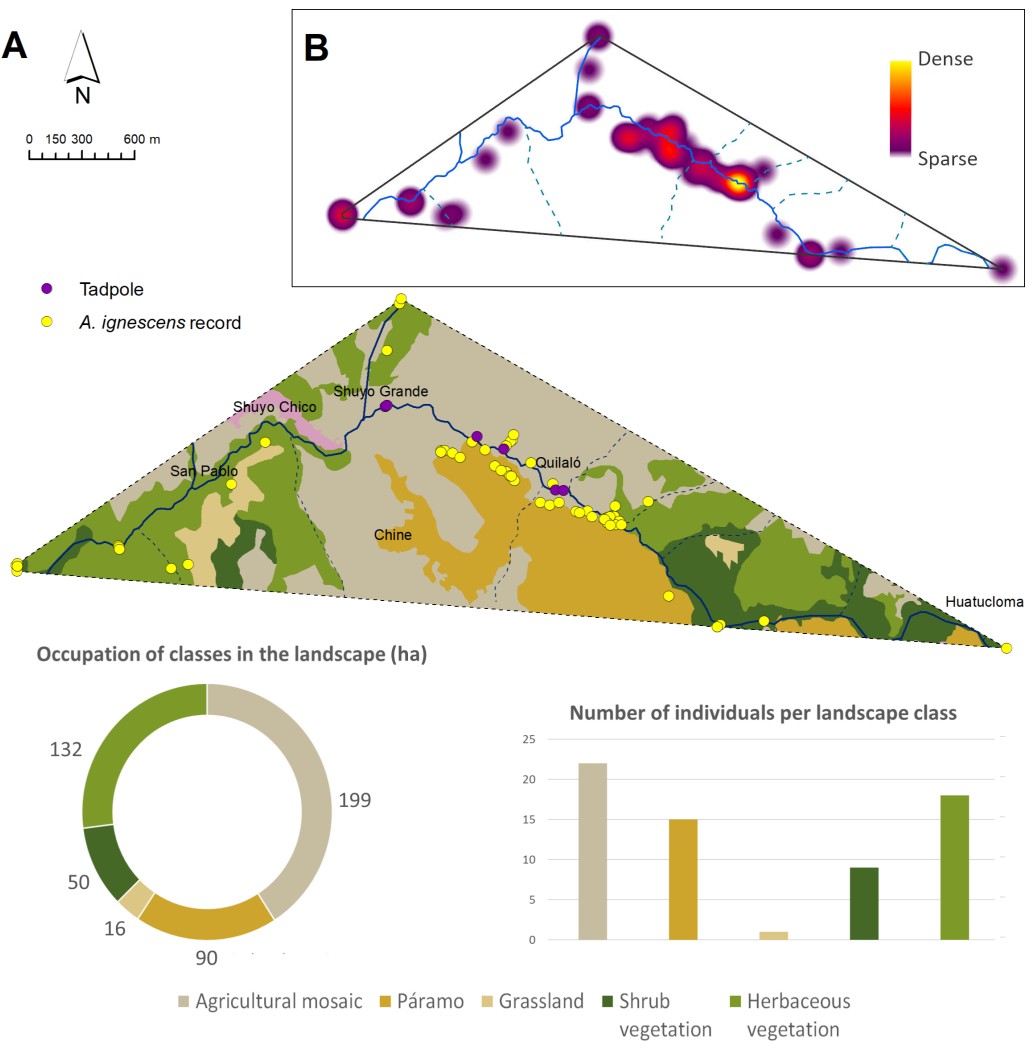

**Figure 5   Occupation of *Atelopus* ignescens.** (A) Extent of Occurrence (EOO) and land use classes, and (B) heat map of records density based on field work. Image prepared by Mateo A. Vega-Yánez.

According to the landscape metrics calculated in the Extent of Occurrence (EOO) of the Jambato (Fig. 5; Table 3), 40.9% of the landscape corresponds to agricultural mosaic which has 10 patches and a patch density of 2.06. Many Jambatos were found walking near a ditch parallel to the Guambaine River, built by the community to water crops (*e.g.*, corn, barley, potato). Also, there was a relatively high density of Jambatos in areas with high grass and cow and sheep excrement. Herbaceous vegetation covers 27% of the surveyed landscape and is composed of 10 patches with a density of 2.06; this habitat is dominated by a native grass of the genus *Pennisetum*. A total of 18 Jambatos were recorded in herbaceous vegetation, including gravid females southeast of San Pablo. Páramos, which in the study area are located between 2,900 to 4,600 m a.s.l., occupy 18.4% of the landscape composed of three patches with a density of 0.62. Páramos suffers from constant anthropogenic pressures to transform the area towards agricultural and cattle farming. Despite being a

**Table 3  Landscape metrics calculated in the extension area of *Atelopus ignescens* presence.**

| Metric | Landscape classes | | | | |
|---|---|---|---|---|---|
| | Agricultural mosaic | Páramo | Grassland | Shrub vegetation | Herbaceous vegetation |
| PLAND (%) | 40.9 | 18.4 | 3.32 | 10.3 | 27 |
| CA | 199 | 90 | 16 | 50 | 132 |
| NP | 10 | 3 | 2 | 5 | 10 |
| PD | 2.06 | 0.62 | 0.41 | 1.03 | 2.06 |
| LPI | 35.5 | 17.3 | 2.89 | 8.27 | 14 |
| DIV | 0.87 | 0.97 | 0.99 | 0.99 | 0.97 |
| AI | 97.4 | 98.1 | 96.2 | 95.6 | 96.6 |
| ENN MN | 108 | 484 | 2456 | 207 | 47 |

Notes.
PLAND, Percentage of landscape; CA, Class Area; NP, Number of Patches; PD, Patch Density; LPI, Largest Patch Index; DIV, Division Index; AI, Aggregation Index; ENN MN, Mean of Euclidean nearest-neighbor distance.

difficult place to search for *A. ignescens* because of its steep slopes and dense grasslands, we found 15 individuals. The shrub vegetation habitat, composed of remnants of native vegetation, has a surface area of 50 ha, and occupies 10.3% of the landscape and has five patches with a density of 1.03; this habitat is composed mainly of different species of ferns, shrubs, mosses, epiphytes, and Araceae. Human intervention there is minimal (*i.e.,* few livestock, crops, and buildings), and we found eight individuals of *A. ignescens* in this type of habitat. Other amphibians found in sympatry with the Jambatos include *Centrolene buckleyi*, *Gastrotheca* aff. *pseustes*, *Pristimantis w-nigrum*, *P. actites* and *Pristimantis* sp. The following reptiles were observed during our study: *Pholidobolus prefrontalis*, *Stenocercus cadlei* and *Saphenophis atahuallpae*.

## Aquatic environments and tadpoles

Streams showed similar water characteristics (Table S2). Saravia stream, and Yalliví and Guambaine rivers at Shuyo, showed presence of fecal coliforms (*Escherichia coli*), and also low riparian quality (low tree cover, deforestation and presence of garbage). However, the Andean Biotic Index showed excellent and good conditions for all sites. The Ucumari River, and one tributary of the Angamarca, passing ''El Shuyo'', showed the best ecological and chemical quality, although no tadpoles or adults have been registered at these sites during the study period. These two sites had less specific conductivity, showing lower dissolved solids in the water, than the other sites that show presence of *A. ignescens*.

Although we sampled numerous streams and rivers, we only found tadpoles (six in total) in the Guambaine River during the months of April, May, June, and October. The search for tadpoles was particularly challenging because of the cold-water temperature (average temperature between 10 and 14 °C at 2,900 m a.s.l.) and the depth of some of the main rivers. All tadpoles found were attached to the base of large sedimentary rocks (Fig. 6) in the rapids of the river; at the time of lifting the rocks, we also noted the presence of algae and several macroinvertebrate species. In tadpoles, body length is 5–8 mm ($7.25 \pm 1.3$) and the tail length is 7–12 mm ($9.5 \pm 1.8$).

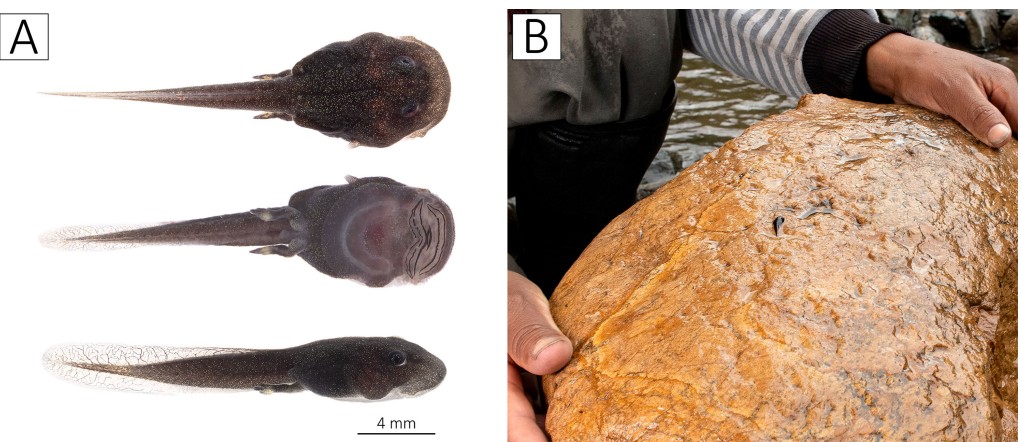

**Figure 6** **Tadpoles of *Atelopus ignescens*.** (A) Tadpole (BL = 8 mm, TAL = 10 mm) in dorsal, ventral, and dorsolateral view, and (B) the flipped rock with a Jambato tadpole, Guambaine River, April 12, 2022. Photos by Amanda B. Quezada-Riera.

## Photo-identification and recaptures

Each Jambato has a unique ventral pattern that allows for reliable photo-identification (Fig. 3). Other useful characteristics include the location of tubercles and spicule on the flanks, and the pattern formed by the capillaries (visible ventrally). Additionally, there is sexual dimorphism in the following traits: (i) females are larger than males; SVL in adult females is 35–57 mm (mean = 43.8 ± 4.8), whereas in adult males is 27–43 mm (mean = 35.1 ± 3.9); (ii) presence of a patch of spiculae in the gular and pectoral region more accentuated in females (*Coloma, Lötters & Salas, 2000*); (iii) in males, presence of nuptial pads on Finger I and II, thickening of forearm and widening of Finger I.

During the year of this study, we only documented two recaptures of Jambatos (adult females), which were first recorded in May 2022, in an area with pastures and shrubby vegetation, close to a small stream that flows into the Angamarca River. Both females presented a grade between 1–2 on the gravidity scale according to *Bronson et al. (2015)*. One of them was recaptured in November and the other in December of the same year, when they presented a grade 3 or 4 on the gravidity scale (Fig. 7).

## Reproduction

We did not find amplectant pairs nor eggs. Evidence that the species is reproducing is the finding of the tadpoles described above. Also, gravid females were found in November and December, February, and May. We found males calling in December, April, and June.

### *Batrachochytrium dendrobatidis*

Our results confirm the presence of the fungus *B. dendrobatidis* (*Bd*) in the Angamarca population. We tested 41 Jambatos for *Bd*, finding 23 positives (12 males and 11 females), during the months of December, April, May, and June (Fig. 8). None of the toads that tested positive showed visible signs of being sick (*e.g.*, abnormal posture, erythema of

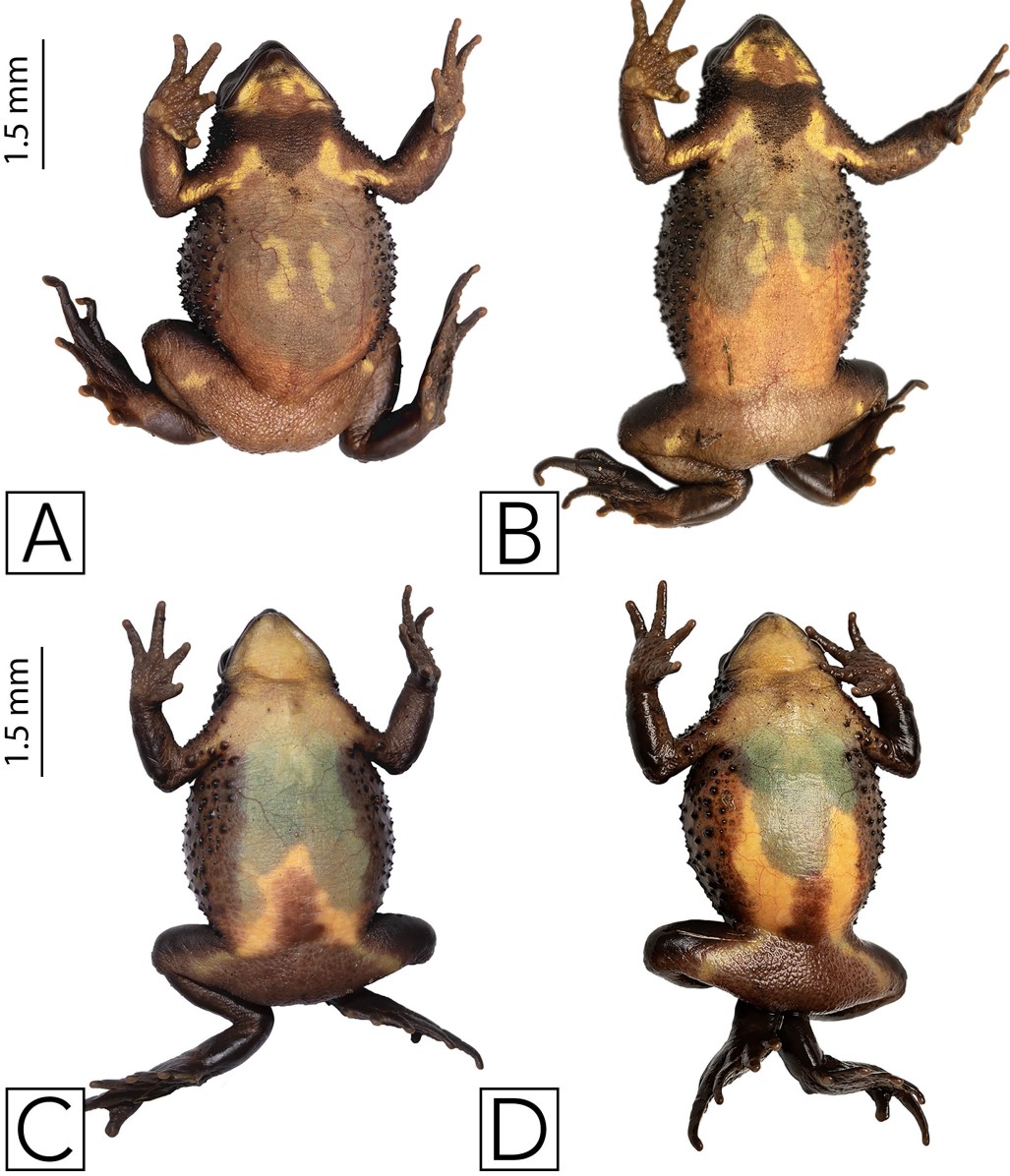

**Figure 7** **Recaptured females of *Atelopus ignescens*, showing different reproductive stages noted by the extent of eggs visible through the ventral skin recaptures and gravidity.** (A) Aig-018 recorded May 5, 2022. (B) Aig-018 recaptured (under Aig-025) November 21, 2022. (C) Aig-021 recorded May 7, 2022. (D) Aig-021 recaptured (under Aig-028) December 16, 2022. Photos by Amanda B. Quezada-Riera.

ventral surfaces, abnormal skin shedding, slow righting reflex, ulceration).We did not test tadpoles for *Bd*.

## Diet of *Atelopus ignescens*

From the obtained fecal samples of *A. ignescens*, a total of 1,263 specimens or identifiable body parts were found, corresponding to 68 different morphospecies in two different phyla, Arthropoda, and Mollusca (Fig. 9; Table S3). Most of the prey consists of arthropods: 46%

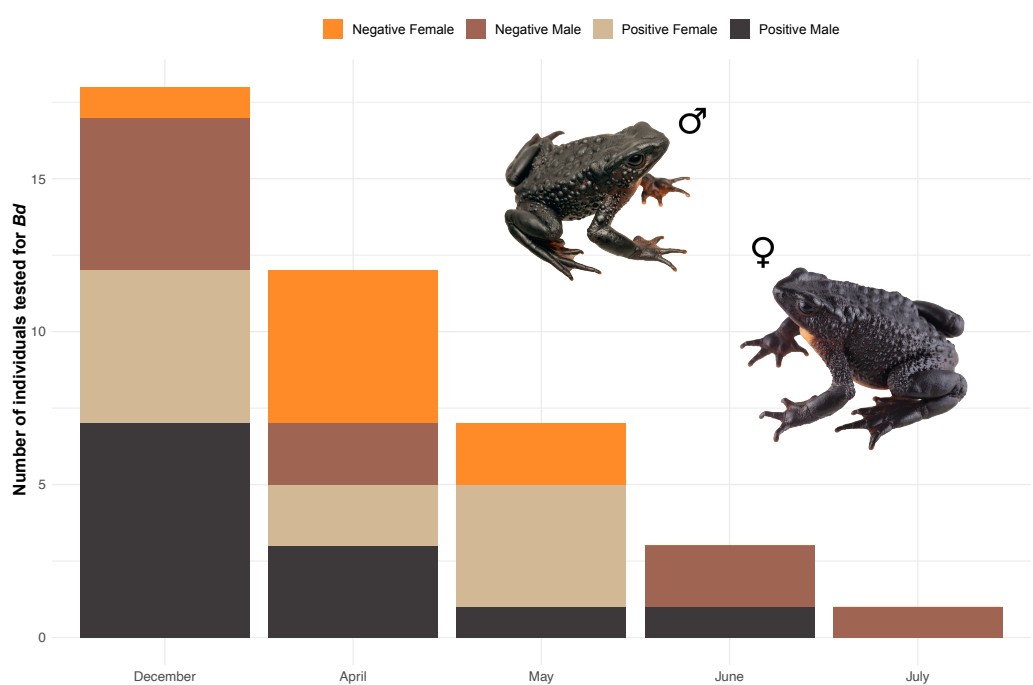

**Figure 8  Stacked bar plot of the 41 Jambatos tested (males and females) for *Bd* in different monitoring months.** Image prepared by Mateo A. Vega-Yánez.

of Acari, 29% of Coleoptera, and 14% of ants. Within Acari, two morphospecies of Oribatid mites were predominant in the feces, morphospecies ECFN5288 (25%) and 5296 (22%) respectively. Within Hymenoptera, one species of Formicidae (ECFN 5234) is clearly predominant (83%), while in Coleoptera two morphospecies, a Staphylinidae (ECFN 5245) with 25% and one species of Latridiidae (ECRN5261) with 24% were prevailing in the feces (Fig. 10).

The morphospecies diversity of Acari is lower than the diversity of Coleoptera, 15 morphospecies of Acari *versus* 30 morphospecies of Coleoptera (Table S4). The mean size of Acari found in the diet is 0.62 mm, and Coleoptera mean size of Elytra is 1.87 mm. Very few complete specimens were found: four complete larvae with mean a size of 3.70 mm, one complete Formicidae specimen (1.98 mm), one Chilopoda (8.46 mm).

## Climate over time and space

Our results show that the year 1983 had the highest temperature overall, and 1975 was the year with the lowest temperature in all localities. In terms of precipitation, 1998 was the year with the highest precipitation and 2005 was the year with the lowest precipitation (Fig. 11). Concerning the average annual temperature and precipitation values for each locality analyzed, the results show that the locality "Jardines Quinta San Agustín Carretera Panamericana Sur km 26 desde Quito a 2 km desde la carretera" has the highest temperature (15 °C) while the locality "Páramo del Antisana" has the lowest temperature recorded

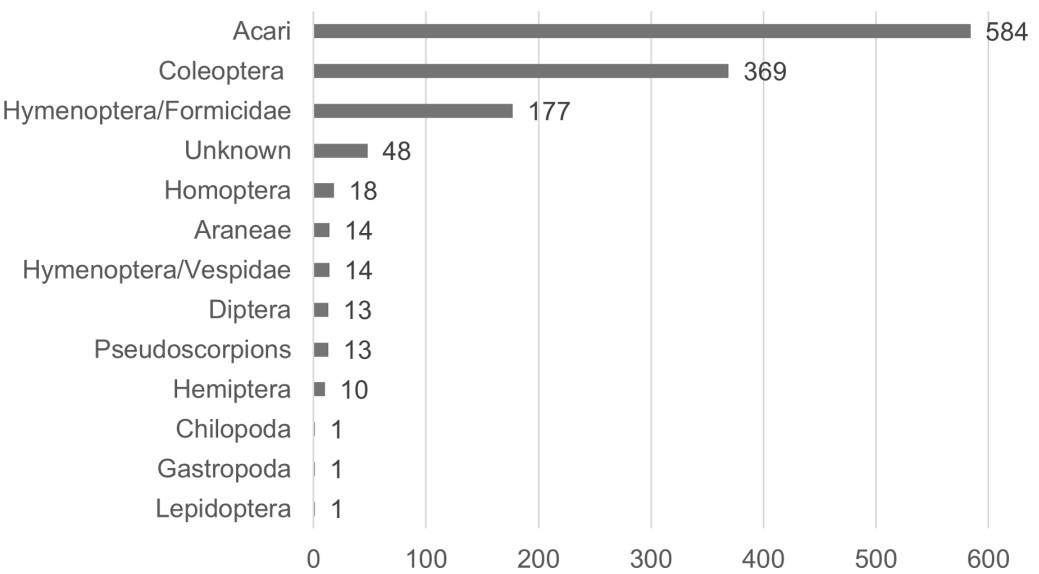

**Figure 9** Diet of *Atelopus ignescens* based on feces from 27 adults found at the Angamarca River basin on May 10th, 2016. Image prepared by Mateo A. Vega-Yánez.

(3.8 °C). In addition, "Papallacta" has the highest precipitation (99 mm) and "Latacunga" has the lowest precipitation value (45 mm) (Table S5).

There are significant differences in temperature and precipitation between historical localities and Angamarca (Table 4). In terms of temperature, only one locality (Papallacta) is similar to Angamarca. For the precipitation analysis, there are three localities ("36.8 km NNE Guaranda", "20 km Norte Riobamba" and "Zumbahua") similar to Angamarca.

Variation of climatic conditions during the analyzed periods (Pre-decline 1960–1980, Population crashes 1981–1990, Current 1991–2021) are shown in Fig. 12. During the Pre-decline period (1960–1980), the average annual temperature was 8.5 °C, with an average variation of 0.44 °C; the average annual precipitation reached 74 mm, with an average variation of 20 mm. During the Population crashes (1981–1990), the average annual temperature barely varied at 8.6 °C, with a decrease in the mean variation to 0.34 °C. The average annual precipitation was 70 mm, with an average variation of six mm. In this period, specifically in 1988, the month with the highest temperature recorded was March (13 °C), coincidentally the last record of Jambato was in that month and year until its subsequent rediscovery. At Current (1991–2021), there is a continuous increase in temperature, reaching an annual average of 8.8 °C, with a mean variation of 0.39 °C. The years 1998, 2015 and 2019 stand out with the highest temperatures. The average annual precipitation increased to 80 mm, with an average variation of 22 mm.

## DISCUSSION

The rediscovery of *Atelopus ignescens* in the Angamarca River basin in 2016 (*Coloma, 2016*), after presumed extinction, is a significant event in the context of amphibian conservation and, more particularly, harlequin toads (*Lötters et al., 2023*). Our study

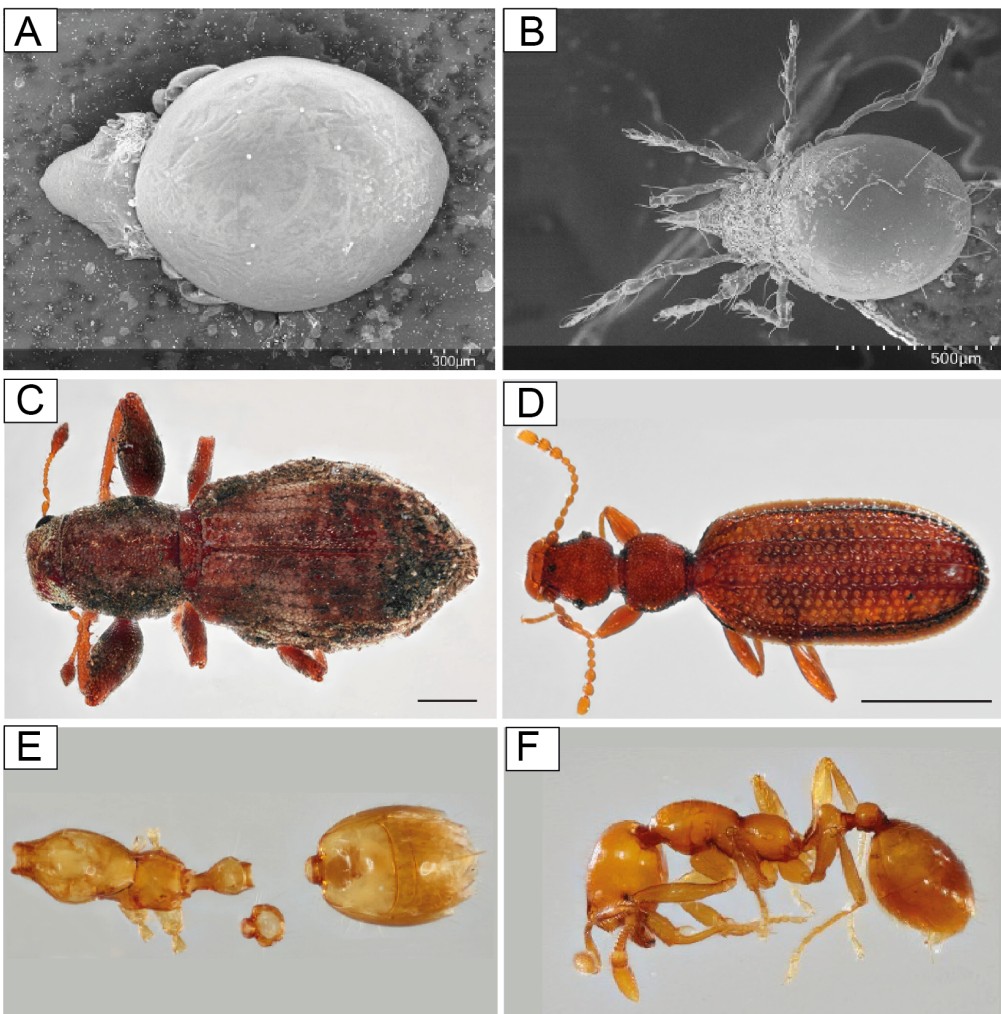

**Figure 10 SEM of Acari from frog feces, Oribatida.** (A) ECFN5288 (B) ECFN5297 Photographs, elytra and abdomen from frog feces and litter, Coleoptera. (C) ECFN10429 (D) ECFN10424 Photographs of ants complete specimens and body parts from frog feces and litter, Formicidae. (E) ECFN5234 (F) ECFN5195. Image prepared by Mateo A. Vega-Yánez.

presents a preliminary overview to understand the current distribution, ecological characteristics, and potential threats faced by this iconic species. We also lay out a series of specific conservation recommendations given what we have learned about the Jambato.

## Occupation of the Jambato in the Angamarca River basin

Our study was intended to have a preliminary understanding of the distribution of the Jambato in its last known locality, the Angamarca River basin. The field surveys reveal marked patterns in the current distribution of *A. ignescens* (based on 71 individuals found in various life stages). Within the basin, the species occupies a range of habitats, including Páramo, shrub vegetation, agricultural mosaic, grasslands, and riverbanks. The extent of the Jambato's presence (EOO) during our monitoring period is 492 ha.

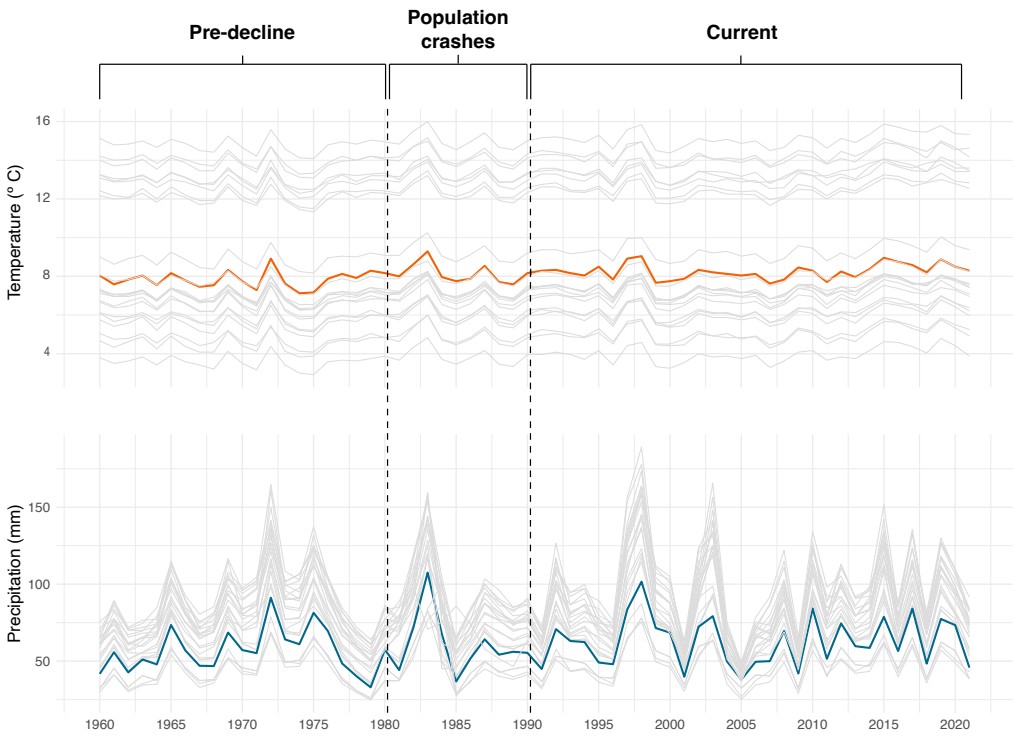

**Figure 11** Average annual temperature and precipitation (1960–2021) at the historical localities (gray lines) of *Atelopus ignescens* and at Angamarca (orange and blue lines). Image prepared by Mateo A. Vega-Yánez.

Interestingly, most Jambatos have been found within the agricultural mosaic (40.9% of the landscape), especially in areas with patches of high grass, ditches, and cow manure, near the Guambaine River (Fig. 5). Although the numbers of Jambatos found in our 12-month survey are encouraging, locals in the Angamarca valley mention that the species used to be extremely abundant in the area, becoming scarce since the late 1980s and early 1990s. Thus, current abundances are lower than the historical presence of the species in the valley.

The diverse habitat use of the Jambato is a challenge for management, highlighting the need to incorporate human-altered landscapes into conservation strategies, but limiting clearly detrimental activities such as burning for agriculture and intense grazing by cattle in areas of relatively high densities of Jambatos.

## Reproduction

While no amplectant pairs or eggs were observed during the study, the presence of gravid females, tadpoles, and calling males indicate reproductive activity that, most likely, is concentrated in the months of December and May. Indirect evidence for reproduction is one of the most relevant results of our study and sheds hope that *in situ* conservation is viable. Our study also identifies the Guambaine River as the only place, so far, suitable for Jambato's tadpoles. We still need efforts to identify the specific sites where adults lay eggs.

**Table 4** Results of the Wilcoxon Rank Sum test (*p*-value < 0.05) of temperature and precipitation between Angamarca and the other localities.

| Code | Locality | Temperature | Precipitation |
|---|---|---|---|
| Locality1 | 20 km Norte Riobamba (carretera Riobamba-Ambato) | 2.2E−16 | 0.07941[*] |
| Locality2 | 36.8 km NNE Guaranda | 2.2E−16 | 0.1256[*] |
| Locality3 | Angamarca | − | − |
| Locality4 | Cayambe | 2.2E−16 | 0.00217 |
| Locality5 | Cráter de Pasochoa | 1.05E−14 | 4.76E−10 |
| Locality6 | Guaranda | 2.2E−16 | 0.0000112 |
| Locality7 | Ingaloma | 2.2E−16 | 7.43E−11 |
| Locality8 | Jardines Quinta San Agustín. Carretera Panamericana Sur km 26 desde Quito A 2 km desde la carretera | 2.2E−16 | 1.54E−13 |
| Locality9 | Laguna de Mojanda | 2.35E−15 | 0.000000135 |
| Locality10 | Laguna La Mica | 2.2E−16 | 0.000000111 |
| Locality11 | Latacunga | 2.2E−16 | 1.83E−08 |
| Locality12 | Limpiopungo | 2.2E−16 | 0.000464 |
| Locality13 | Machachi | 2.2E−16 | 3.82E−09 |
| Locality14 | Papallacta | 0.1505[*] | 2.61E−15 |
| Locality15 | Páramo de Guamaní | 2.2E−16 | 5.58E−10 |
| Locality16 | Páramo del Antisana | 2.2E−16 | 0.0002193 |
| Locality17 | Pie del volcán Chimborazo, vía Ambato - Guaranda | 2.2E−16 | 0.0000072 |
| Locality18 | Pintag | 2.2E−16 | 1.01E−14 |
| Locality19 | Quijos-Oyacachi | 2.2E−16 | 7.29E−13 |
| Locality20 | Quito | 2.2E−16 | 5.05E−13 |
| Locality21 | Río Cutuchi | 2.2E−16 | 0.00000422 |
| Locality22 | Vía Salcedo-Oriente | 1.02E−14 | 1.18E−10 |
| Locality23 | Zumbahua | 2.2E−16 | 0.07161[*] |

**Notes.**
Values with an asterisk (*) are similar to Angamarca.

## Diseases

Some of the first reports of the chytrid fungus in South America come from *Atelopus* collected during the late 1980s (*Ron & Merino, 2000*; *Bonaccorso et al., 2003*). According to locals, this is the same time period when Jambatos became rare in the Angamarca basin. The impact of chytridiomycosis in amphibians declines and extinction is well documented worldwide (*e.g.*, *Scheele et al., 2020*), and has been particularly catastrophic in harlequin toads (*La Marca et al., 2005*; *Lötters et al., 2023*). Our confirmation of *Bd* in the Angamarca population raises concerns about the future of the species. However, although the prevalence of *Bd* disease in the Jambato population is relatively high (32%), the population appears to be healthy.

Angamarca represents a unique opportunity for studying the dynamics between *Bd* and *Atelopus ignescens*, one of the last survivors of the genus in the Ecuadorian Andes (*Lötters et al., 2023*). Two main hypotheses explain the persistence of the Jambato in the presence of *Bd*: (i) A less virulent strain of *Bd* or (ii) a higher resistance to the disease, provided either by the Jambato's microbiome, skin toxins, and/or its immune system. Determining which

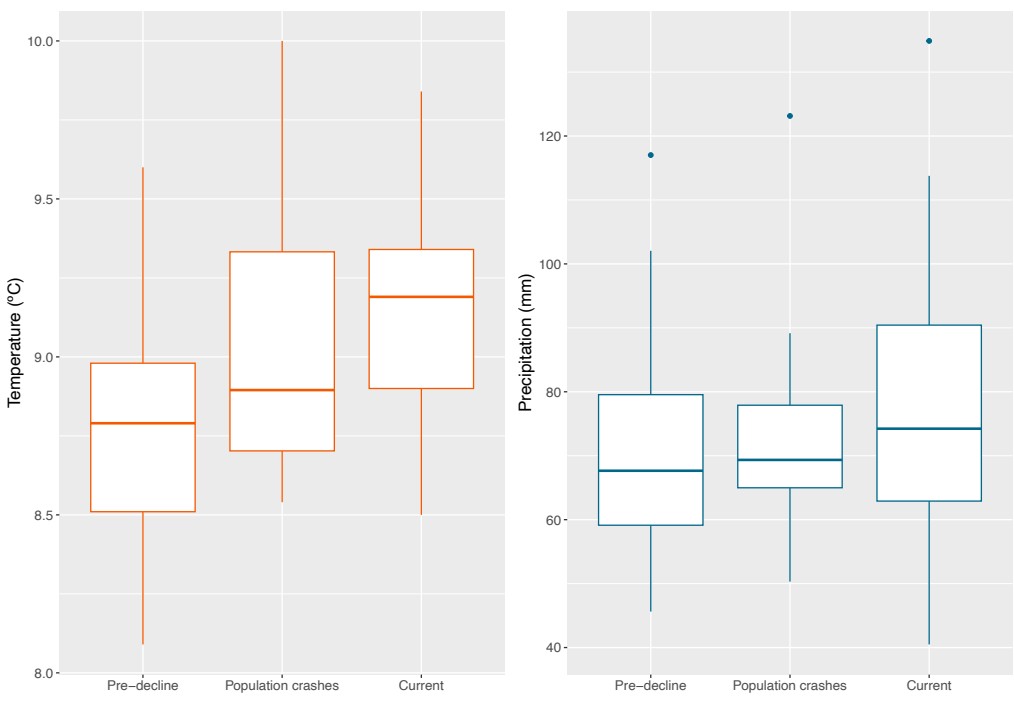

**Figure 12** Boxplot showing the climatic conditions of all Jambato's localities (historical and present) during the analyzed timeframes (Pre-decline 1960–1980, Population crashes 1981–1990, Current 1991–2021). (A) Average annual temperature. (B) Mean annual precipitation. Image prepared by Mateo A. Vega-Yánez.

of the two hypotheses is correct might hold the key to the conservation of the Jambato and other endangered amphibian species.

As documented, the Jambato population in Angamarca faces multiple threats at the same time (high temperatures, drought, invasive species, fires). In theory, stressors might weaken the Jambato's defenses against disease (*Rollins-Smith & Le Sage, 2023*). Therefore, the Jambato remains a remarkable example of resistance and deserves further study. Additionally, further studies should also concentrate on the dynamics of *Bd* in the amphibian community in Angamarca.

## Climatic analysis

It is difficult to elucidate how climatic variables have affected the abundance of *A. ignescens*. Over 61 years of analysis at Angamarca and other historical localities, there is evidence of an increasing trend in temperature. In addition, our analyses suggest that between 1991 and 2021 there has been an increase in the variation of precipitation patterns, both in minimum and maximum values. The combination of warm temperatures and low precipitation, especially in 1987, has been suggested as one of the factors contributing to the population declines of the Jambato (*Ron et al., 2003*), most likely in synergy with the introduction of *Bd* (see previous section).

Warm temperature by itself is insufficient to explain declines; according to *Pounds et al. (2006)* about 80% of the species that have disappeared were last sighted after relatively warm years and combined data point to the decline of the most vulnerable species towards the end of the 1980s. For example, our analyses show that the hottest year on record was 1983, but population crashes were not observed in Angamarca until the late 1980s, according to locals; however, it is possible that the species may have slowly declined as a result of higher adult mortality and low reproductive success. Likewise, when analyzing precipitation patterns during the pre-decline and population crashes periods (1960–1990), the driest year was 1979, again several years before observed declines. Thus, given these data, a most likely scenario explaining the Jambato declines is associated with the arrival and dispersal of *Bd,* as observed in other species across the globe (*Scheele et al., 2020*).

Undoubtedly, it is necessary to determine the changes in temperature and precipitation patterns related to climate change in amphibians, considering that the effects might be diverse (*Wake, 2007*; *Luedtke et al., 2023*). Increasing temperatures can alter microbial communities in the skin and digestive system of amphibians impairing their ability to resist pathogens (*Rollins-Smith & Le Sage, 2023*). We consider it relevant to further understand this relationship. Also, it is important to assess other climatic variables such as ultraviolet B radiation (UVBR), which in the tropical Andes (close to latitude 0) might have greater effect than in other regions, increasing vulnerability to diseases (*Cramp & Franklin, 2018*; *Lundsgaard, Cramp & Franklin, 2020*).

## Aquatic ecosystems

Although human disturbance in the Angamarca River Basin is conspicuous (*i.e.,* agriculture, cattle, garbage), water quality remains high at all sites, as measured by the Andean Biotic Index. We note that the flow of nutrients used in agricultural activities might be modifying the chemical dynamics of streams and rivers. Although some of these changes might be negative for the aquatic fauna, an increase of phosphorus and nitrogen might raise the availability of algae, a key resource for Jambatos tadpoles. We also documented the presence of fecal coliforms in some streams and rivers (Saravia stream, Yalliví and Guambaine rivers), representing a health risk to the human population inhabiting the basin.

During our surveys in the Angamarca basin, when searching for tadpoles, we observed the presence of the introduced Rainbow Trout (*Onchorynchus mykiss*) in most of the streams and rivers of the basin. Recent studies have demonstrated that the Rainbow Trout has a negative effect on Andean amphibians (*Martín-Torrijos et al., 2016*; *Krynak et al., 2020*) and invertebrate communities (*Vimos et al., 2015*). Although no specific studies have been performed at the Angamarca basin, it is probable that Jambatos (eggs, tadpoles, juveniles, and adults) are also preyed on by this exotic trout. Under a strict conservation scenario, the eradication of the Rainbow Trout would be the obvious solution; however, trout is an important food source for the human communities of the Angamarca valley. Therefore, as a first management step we recommend trout extirpation from areas where tadpoles have been found (Guambaine River, near Quilaló). Also, we have identified areas with relatively high densities of calling males and gravid females; streams and rivers nearby these areas should also be trout-free.

## Diet

Herein we present the first evaluation of the diet of the Jambato, based on feces analysis content. Identification of fragmented body parts in the feces is not always an easy task. Therefore, collecting litter samples from the same locality is a useful method to help matching and identifying organisms found in the feces content. For example, the Oribatid mite (ECFN 5282) from the feces corresponds to the mite (ECFN 950) from the litter, same for the mite (ECFN 5299) found in the feces that corresponds to the mite (ECFN 982). For other organisms such as beetles and Homoptera, the method is even more useful, since most of the animal is digested, and only the elytra or hard plate remains in the feces which are difficult to identify by themselves. For example, the litter sampling at the same locality enabled us to match the elytra (ECFN 5236) to a Carabid beetle (ECFN 10429), as well the elytra (ECFN 5254), was paired to the Coleoptera (ECFN 10424) found in the litter sample.

Based on our result, *A. ignescens* feeds on small arthropods, with a diet composed predominantly of Oribatid mites, Coleoptera, and ants. Stomach-gut analyses are more broadly used to evaluate toad diets, considering that fecal samples can lead to an underestimation in the number and diversity of prey taxa, due to the complete digestion of small and fragile animals. The downside of the method is the invasive process of flushing the gut content that negatively affects the animal. In the case of rare, critically endangered species such as *A. ignescens*, the stomach-gut method was considered too invasive, while the fecal composition presented a safe method for a first assessment of the diet. Interestingly, our results show that *A. ignescens* primary source of food extracted from the feces are Oribatid mites. Poison frogs (*e.g., Oophaga*) are also specialized in ants and mites (*Caldwell, 1996*; *Darst et al., 2005*; *Mcgugan et al., 2016*). Thus, it is possible that skin toxins found in *Atelopus* (*e.g.*, guanidinium alkaloids; *Pearson & Tarvin, 2022*) might have a dietary origin.

## Recommendations for the conservation of the Jambato

The data presented in this study is a first step for more in-depth conservation actions. As an important advance into this direction, the Alianza Jambato Foundation (AJF) was recently formed (https://alianzajambato.org/). Members of the AJF have already established a series of activities at Angamarca. We are very much aware that the Jambato population is still in fragile condition and a monitoring program is now in place. Paralelly, based on the results of this study, we recommend the following actions:

## Terrestrial landscape management

Our preliminary study shows the Jambato's occupation patterns (Fig. 5). The habitat management of these areas is an important first step for the long-term *in situ* conservation of the Jambato. Specifically, we recommend the following activities: (i) Limiting burning in areas where Jambatos are more commonly found, (ii) limiting the access of cows to the areas with high density of harlequin toads, (ii) increasing habitat heterogeneity (hiding places such as rocks and tall grass), (iii) maintaining sources of food (cow manure increases the number of insects in the area), (iv) avoiding constructions and solid fences

between terrestrial landscapes with high density of adults and reproductive sites (rivers and streams). An additional conservation action for the Jambato could be the reintroduction of laboratory-bred individuals (currently at Centro Jambatu: http://www.anfibiosecuador.ec/), although we are aware that this represents a great challenge in terms of the survival rate of reintroduced amphibians (see *Klocke et al., 2023*). Similarly, in order to extend the range of distribution of the Jambato in Angamarca, translocations should be considered (in areas with similar environmental conditions); however, this step must follow successful management in the current locality with a documented increase of the population.

## Aquatic landscape management

A first important measure is the extirpation of Rainbow Trout from rivers where there is a high density of Jambato tadpoles and reproductive sites. With the information at hand, we suggest that the extirpation initiatives should be concentrated in the Guambaine River, neary Quilaló. Although there is not much information on this type of extirpation, examples from small watersheds, like the Guambaine, the extirpation of the Rainbow Trout has been successful to recover native aquatic populations in Japan (*Morita, 2022*). Other important measures include riparian area protection, to prevent the pollution with fecal coliforms, pesticides and fertilizers used in the agro-cattle mosaic (*Dunn et al., 2022*; *Graziano, Deguire & Surasinghe, 2022*). Preserving at least 50 m wide of riparian vegetation has demonstrated to be effective in the conservation of most aquatic sensitive groups in streams and smaller rivers in the Neotropics (*Dala-Corte et al., 2020*), as is the case of the rivers in the Angamarca region. This is important also to protect the tadpoles and the adults found in the surroundings of aquatic environments. The areas that concentrate most of the human population are also where most fecal coliforms are detected. Therefore, to protect the local population and the aquatic environments that they depend on, water treatment plants should be put in place. Small examples like wetlands or vermifilters can be very effective at this scale (*Chowdhury, Bhunia & Surampalli, 2022*). This will also reduce the chances of introducing pathogens to the aquatic environments that can affect tadpoles.

## Local community involvement

This research was born as a result of a previous socio-environmental research that identified gaps between the community and scientists, but also opportunities to start an in-situ conservation project (*Vizcaíno-Barba et al., 2022a*). One of the main actions developed concurrently with the present study was the active involvement of local communities parallel to research activities. Because the Jambato primarily inhabits agricultural areas, conservation efforts must have a community-based focus. As a result of these actions, the Alianza Jambato was born, an initiative that has brought together approximately 50 individuals, including professionals from various fields and 26 national and international institutions, and members of the communities of Angamarca (*Vizcaíno-Barba et al., 2022b*).

One of these institutions is the Parish Government of Angamarca, with whom we closely collaborate to develop effective conservation policies. This interdisciplinary and coordinated work has allowed the people of Angamarca to become proud of the Jambato toad and interested in its conservation. This small toad had little significance to the

community, but now the Jambato even has its own celebration day in Angamarca (April 21st). This resolution includes measures to preserve the Jambato's habitat, arising from the current study and the community outreach work carried out. Continuing the involvement of the Angamarca community is at the core of all the other conservation and management activities mentioned in this last section.

## CONCLUSIONS

Angamarca is the last known locality of *Atelopus ignescens* where we have recorded during one year of monitoring 71 individuals in different stages of development. The extent of Jambato's presence (EOO) is 492 ha where the largest number of individuals found has been in agricultural mosaic, and in less intervened areas with a higher cover of shrub vegetation we have recorded few individuals. Meanwhile, we found six tadpoles in the Guambaine River and in places with the best ecological and chemical quality, although no tadpoles have been registered at these sites during the study period. We tested *Bd* for 41 Jambatos, finding 23 positives (12 males and 11 females), none of the individuals that tested positive had visible signs of being sick. It is a challenge to conserve the species in such a complex landscape, but we emphasize the importance of managing both terrestrial and aquatic landscapes for the long-term conservation of this species. It is also important to involve the community that coexists with the species.

## ACKNOWLEDGEMENTS

Special thanks to the Angamarca community, and to Andrés Morabowen (UDLA) who provided indispensable field and lab assistance related to the assessment of rivers.

### Funding

Funding was provided by the Amphibian Survival Alliance, Stiftung Artenschutz, the Atelopus Survival Initiative, Re:wild, Auckland Zoo, the Indianapolis Zoo, the Jocotoco Foundation, Universidad Tecnológica Indoamérica, the BIOMAS Research Group (UDLA), and Universidad San Francisco de Quito USFQ (HUBI 5467, 16871, 17857). The funders had no role in study design, data collection and analysis, decision to publish, or preparation of the manuscript.

### Grant Disclosures

The following grant information was disclosed by the authors:
Amphibian Survival Alliance, Stiftung Artenschutz, Atelopus Survival Initiative, Re:wild, Auckland Zoo, Indianapolis Zoo, Jocotoco Foundation, Universidad Tecnológica Indoamérica, BIOMAS Research Group (UDLA), and Universidad San Francisco de Quito USFQ: HUBI 5467, 16871, 17857.

### Competing Interests

The authors declare there are no competing interests.

## Author Contributions

- Mateo A. Vega-Yánez conceived and designed the experiments, performed the experiments, analyzed the data, prepared figures and/or tables, authored or reviewed drafts of the article, and approved the final draft.
- Amanda B. Quezada-Riera performed the experiments, analyzed the data, prepared figures and/or tables, authored or reviewed drafts of the article, and approved the final draft.
- Blanca Rios-Touma performed the experiments, analyzed the data, prepared figures and/or tables, authored or reviewed drafts of the article, and approved the final draft.
- María del Carmen Vizcaíno-Barba conceived and designed the experiments, performed the experiments, analyzed the data, authored or reviewed drafts of the article, and approved the final draft.
- William Millingalli performed the experiments, authored or reviewed drafts of the article, and approved the final draft.
- Orlando Ganzino performed the experiments, authored or reviewed drafts of the article, and approved the final draft.
- Luis A. Coloma performed the experiments, analyzed the data, prepared figures and/or tables, authored or reviewed drafts of the article, and approved the final draft.
- Elicio E. Tapia performed the experiments, analyzed the data, authored or reviewed drafts of the article, and approved the final draft.
- Nadine Dupérré performed the experiments, analyzed the data, prepared figures and/or tables, authored or reviewed drafts of the article, and approved the final draft.
- Mónica Páez-Vacas performed the experiments, authored or reviewed drafts of the article, and approved the final draft.
- David Parra-Puente conceived and designed the experiments, performed the experiments, authored or reviewed drafts of the article, and approved the final draft.
- Daniela Franco-Mena performed the experiments, analyzed the data, prepared figures and/or tables, authored or reviewed drafts of the article, and approved the final draft.
- Gabriela Gavilanes performed the experiments, analyzed the data, authored or reviewed drafts of the article, and approved the final draft.
- David Salazar-Valenzuela performed the experiments, authored or reviewed drafts of the article, and approved the final draft.
- Carlos A. Valle performed the experiments, authored or reviewed drafts of the article, and approved the final draft.
- Juan M. Guayasamin conceived and designed the experiments, performed the experiments, analyzed the data, prepared figures and/or tables, authored or reviewed drafts of the article, and approved the final draft.

## Field Study Permissions

The following information was supplied relating to field study approvals (i.e., approving body and any reference numbers):

Our research was carried out with the permission of Ecuadorian Ministerio del Ambiente, Agua y Transición Ecológica (MAATE): (MAATE-DBI-CM-2022-0245).

## Data Availability

   The data is available in the Supplementary Files.

## Supplemental Information

Supplemental information for this article can be found online at http://dx.doi.org/10.7717/peerj.17344#supplemental-information.

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
