# Peer review of "Path for recovery: an ecological overview of the Jambato Harlequin Toad (Bufonidae: Atelopus ignescens) in its last known locality, Angamarca Valley, Ecuador"

_PeerJ, doi:10.7717/peerj.17344_

## Round 0.1 · original submission · Major Revisions

Dear authors,

The major decision has been based on the number of observations made by the referees rather than on the implications of their comments on the contents and structure of your work.

It is up to you to consider the call description in a separate manuscript or to keep it in this work.

I look forward to hearing from your revised version

Emmanuel

Reviewer 1 ·

Basic reporting

This manuscript is written in clear and unambigous professional english througout. I very much enjoyed the historic background, especially some of the old records of Jiménez de la Espada etc. in the introduction. This is a complex story with a potential extinction and a rediscovery, masterfully narrated in the introduction.
Materials and methods are comprehensive and detailed. The description of the sites and habitats, The only part that seems a bit weak is the analysis where there is a general statement at the end 'All analyses were performed in R version 4.3.1.' but what packages where used? How did you analyse the data for each one of this multidisciplinary paper? Or is it all mostly descriptive?
Results: I'm aware there is not really a case here to create an SDM for the species in this context, but what about a heatmap in qGIS? IT would tell you more about the distribution of the points and perhaps aviod having to reveal the exact location of a toad that might be sought after in the pet trade (e.g. https://www.qgistutorials.com/en/docs/3/creating_heatmaps.html).
Climatic analysis: If you were trying to interpret this in the lense of the Pounds et al.2006 paper, why not do so explicitly? The hypothesis was more complext though than locally warm years. I am not a Bd sceptic, but where is the data to support this statement? "Thus, given these data, a most likely scenario explaining the Jambato declines is associated with the arrival and dispersal of Bd." Similarl ideas have been proposed for the east coast of Australia, but the data was simply not there. https://journals.plos.org/plosone/article?id=10.1371/journal.pone.0052502, and in other cases the links where there but the timing wasn't that clear https://royalsocietypublishing.org/doi/10.1098/rspb.2013.1290, that paper has all the r code to replicate what has been done and explore the ecuadorian data in a similar spatio-temporal context that could be easily replicated (I acknolwedge not for this paper). Perhaps what I'm asking her, don't oversimplify something without presenting clear evidence for this. With regards to conservation, would translocation be a useful tool to help it expand its range? Is it pointless in this case?

Experimental design

The paper has an original approach within aims and scope for the paper. It uses a multidisciplinary approach to describe the status, biology and distribution of this species, which was previously thought to be extinct. The research question are presented clearly, and the methods are describe with enough detail that it could be replicated.

Validity of the findings

The manuscript deals mostly with the natural history of a critically endangered and recently thought to be extinct of harlequin toad, which is an important contribution in an era where many species in this genus seem to be coming back. It certainly gives the backbone for further studies, which could untanble their incredible comback and coexistence with an emerging disease that caused their mass extinction in the first place. The conclusions are well stated, I would appreciate more on the conservation efforts, that given the authors experience they would recommend as stated above. Is this species now on solid footing? Should we try to be expanding its range artifically or is natural dispersal going to be enough for it to recover?

·

Basic reporting

No comment.

Experimental design

No comment.

Validity of the findings

No comment

Additional comments

The paper ”Path for recovery: an ecological overview of the Jambato Harlequin Toad (Bufonidae: Atelopus ignescens) in its last known locality, Angamarca Valley, Ecuador“ by Vega-Yánez et al. deals with conservation-related research on one of the most threatened – and most enigmatic – amphibians from Andean Ecuador. The paper provides important life history data for in situ and ex situ conservation measures, both in process for a couple of years already. I am fine with the questions asked in this paper and not aiming at hypothesis testing.
As an option, I would like to ask the authors to consider to exclude the call description, as there is next to no direct benefit of this information for conservation practices (different to climate, food, Bd etc.). The call description may well go in a separate short note, making the species even more visible.
Apart from this, I have minor comments or recommendations only and I hope the authors find them useful.
I like the paper, find it informative with very cool illustrations, useful and well-done. I suggest minor revision.
Stefan Lötters

Comments
Title: Here you refer to “harlequin toad”, while elsewhere in the text you refer to “harlequin frog” – never “harlequin toad” but you use “toad”. Chose one, both is possible but not within the same piece of text I would say.
Line 58: I suggest removing key words that appear in the title.
Line 67: Say “2,800 to 4,200 m a.s.l.”
Line 69: Provide full name of Jimenez de la Espada, which reads better, I think; otherwise place year and page after mentioning the name here and delete at the end of the sentence.
Line 71: What does he describe with regard to reproductive habits?
Line 72: Rewrite: “Years later, in 1981, at the same site…”.
Line 86: Move year in parentheses behind name and delete at the end of the sentence.
Line 87: Say “public monument”, if true.
Line 89: “last record” not correct, restate like “last record for almost three decades” or so.
Line 90: Be precise, what do you mean by “prior”?
Line 94: Although La Marca (2005) is the first genus-wide study, the update of that study (Lötters et al. 2003) refer to more species (131 instead of 113) – including those undescribed or not identified in 2005 – and provides data on declines. Maybe cite both here.
Line 97: Say “amphibian skin fungus”.
Line 103: Say “pathogenic fungus”.
Line 122: I recommend to exclusively or in addition mention the full Spanish name of the ministry.
Line 126: Place year in parentheses.
Line 128: Say “2,812 m a.s.l.”
Line 135: Say “4,064 m. a.s.l.”
Line 138: You refer to WorldClim2 data here that are based on interpolations. You should refer to this circumstance, as interpolations in the Andes may easily have errors. Also mention the period for with these data are valid. See below my comment on line 259.
Line 139: You have super nice and informative illustrations including ventral aspects of the jambato. It would be informative to the reader to see what the species looks like from above (I mean the adult, not only the tadpole that you illustrate). Maybe add a specimen on white background to figure 1 here. Alternatively, maybe right in the beginning show a separate illustration of the frog in its natural habitat (a new figure 1), which provides a better impression of the study organism to the reader who is not so much into Atelopus.
Line 140f.: I think that he term “opportunistic” should be added somewhere in this paragraph, as this is a common term for how data were obtained.
Line 147: How many nocturnal surveys, at which hours? Rephrase “with the goal to find sleeping individuals”. What about nocturnal activity of tadpoles? Did you look for tads at night?
Line 162, line 226: Which type of gloves?
Line 167: Instead of “categories” better refer to the common term “landuse class” throughout the paper.
Line 168f.: What is a “spatial spread of areas”? You mean the spread of specimens within the area?
Line 170: Not “1 km2” but “1 km²”. BTW; this is quite coarse, given the general size of the area. Couldn’t you downscale?
Line 178: What do you mean by “two lacking any presence record”? Please rewrite in way that you have not recorded the here. Maybe state how far the nearest ever captured specimen was to provide a better picture of apparent absence.
Line 200: Amanda analyzed the photographs by visual inspection only? No software was used? You could easily provide them all in an electronic appendix here.
Line 205: Say “2,971 m a.s.l.”
Line 242: Add “a.s.l.” again to the elevations mentioned here.
Line 259: Climate is defined as the mean weather conditions of the atmosphere over 30 years at a given site. I doubt the 51-year-period (1960-2021) given here for WorldClim 2.1. According to https://www.worldclim.com/version2, this period is 1970-2000.
Line 265: I am not an expert and maybe I am wrong here. The longer the periods are, the more unnormalities are smoothened. Are the results the same when comparing e.g. only 1971-1980, 1981-1990, 1991-2000?
Line 273: Instead of “recognized as different” say “considered to be likely different”. BTW; I find this tricky, as, given figure 2, all 6 larvae were found over a distance of 1,000 m in the same stream and tadpole symbols in the figure even overlay suggesting they were close to each other. Honestly, we don’t know much about larval movement or drift in Atelopus.
Line 276: Maybe add to “habitat” the term “landuse classes” in parentheses (see comment to line 167). This also applies to the text of figure 5 and to line 407f (and perhaps elsewhere in the text, figure legends, tables, appendices).
Line 294: What is “minimal”?
Line 299f.: You do not refer to figure 6A in the text.
Line 323: As you also did the photo ID method to learn more about recaptures (cf. title of paragraph!) you state “our study was not designed with the aim of recapturing individuals”. This is a bit unlucky, although I understand what you mean. However, this should be rewritten. I would especially recommend to emphasize that during the entire period you ONLY had 2 recaptures, which is remarkable.
Line 328f: You mentioned above (MM) where photographs are deposited. Do not repeat here.
Line 353: I there seasonal variation? Worth checking and mentioning. Maybe plot time against number of positives/negatives and show. Tadpoles were not tested for Bd, correct? Maybe mention in MM if not there.
Line 354: What kind of signs of sickness would you expect? What did you look for? Maybe also mention in MM.
Line 380: Maybe consider to show Table S6 in the main paper, not the appendix. I find it interesting.
Line 383f.: This is what I mentioned before. There should be an effect of length of period, too!
Line 430: use the abbreviation “Bd” here.
Line 432: Is this prevalence among all A. ignescens samples over time? What about seasonal variation? What about prevalence of the entire amphibian community? Be clear here.
Line 435: “Lötters” not “Lotters” please, it makes a difference in German language.
Line 441: It is not clear for all the stressors mentioned why they affect A. ignescens in Angamarca.
Line 454: Maybe. But one may also see a direct link between the 1983 warm period and declines some years later. First, there is no robust data. What is ‘the end of the 1980s’? Maybe it only then became more obvious to the people, as probably the species silently started to decline before. Second, what about delayed effects, e.g. a short life span combined with lack of reproductive success?
Line 692: “Köhler”, not “Köehler” please.

---

## Round 0.2 · accepted · Accept

Congratulations, I am so happy to see you have addressed all the comments made by the reviewers.

---

## Author Rebuttal · Round 0.2

**Dear Dr. Emmanuel Serrano**

**Academic Editor**
According to the comments and suggestions proposed by you and the reviewers, we have made corrections to the manuscript "Path for recovery: An ecological overview of the Jambato Harlequin Toad (Bufonidae: *Atelopus ignescens*) in its last known locality, Angamarca Valley, Ecuador" by Vega-Yánez and collaborators.

We appreciate the attention and interest shown in our MS and consider that all the reviewers' suggestions and changes have been incorporated. We have also complied with the suggestion to exclude the description of the call of the MS. All the responses and changes are detailed below in blue.
* * *
Manuscript **"Path for recovery: An ecological overview of the Jambato Harlequin Toad (Bufonidae: *Atelopus ignescens*) in its last known locality, Angamarca Valley, Ecuador"** by Vega-Yánez et al.

**Reviewer 1 (Anonymous)**

**Basic reporting**

This manuscript is written in clear and unambiguous professional English throughout. I very much enjoyed the historic background, especially some of the old records of Jiménez de la Espada etc. in the introduction. This is a complex story with a potential extinction and a rediscovery, masterfully narrated in the introduction.

Materials and methods are comprehensive and detailed. The description of the sites and habitats, The only part that seems a bit weak is the analysis where there is a general statement at the end 'All analyses were performed in R version 4.3.1.' but what packages where used? How did you analyze the data for each one of this multidisciplinary paper? Or is it all mostly descriptive?

R: We appreciate the positive comments by the reviewer!

We have included the name of the R packages used for the analyses. For the climatic part we carried out a non-parametric test (Wilcoxon rank-sum test) to see the differences between temperature and precipitation between Angamarca and the historical localities of the Jambato (see line numbers 246–248). We complemented these analyses with boxplots to visualize the data (see Figure 12).

Results: I'm aware there is not really a case here to create an SDM for the species in this context, but what about a heatmap in qGIS? IT would tell you more about the distribution of the points and perhaps avoid having to reveal the exact location of a toad that might be sought after in the pet trade (e.g. https://www.qgistutorials.com/en/docs/3/creating_heatmaps.html).

R: It is true that due to the quality of the data it is not possible to perform an SDM. However, we have complied with the reviewer's suggestion to create a heat map of the Jambato records in Angamarca. The new figure has been included in the MS (see Figure 5).

Climatic analysis: If you were trying to interpret this in the lense of the Pounds et al.2006 paper, why not do so explicitly? The hypothesis was more complex though than locally warm years. I am not a Bd sceptic, but where is the data to support this statement? "Thus, given these data, a most likely scenario explaining the Jambato declines is associated with the arrival and dispersal of Bd." Similarly ideas have been proposed for the east coast of Australia, but the data was simply not there. https://journals.plos.org/plosone/article?id=10.1371/journal.pone.0052502, and in other cases the links where there but the timing wasn't that clear https://royalsocietypublishing.org/doi/10.1098/rspb.2013.1290, that paper has all the r code to replicate what has been done and explore the ecuadorian data in a similar spatio-temporal context that could be easily replicated (I acknowledge not for this paper). Perhaps what I'm asking her, don't oversimplify something without presenting clear evidence for this.

R: Thanks for the comment! We have included an explicit reference to Pounds et al. (2006) in the discussion. As the reviewers suggests, the scenario associated to the declines of Atelopus *ignescens* might be more complex (and worth analyzing in another paper). We do consider, however, that the abrupt declines observed in the Jambato, and the lack of temporal matching with warm and dry years, are more in agreement with a disease related scenario (line numbers 426–432).

With regards to conservation, would translocation be a useful tool to help it expand its range? Is it pointless in this case?

R: We have incorporated the suggested change. See line number 506–509.

**Experimental design**

The paper has an original approach within aims and scope for the paper. It uses a multidisciplinary approach to describe the status, biology and distribution of this species, which was previously thought to be extinct. The research question are presented clearly, and the methods are describe with enough detail that it could be replicated.

R: Thanks for your positive comments!

**Validity of the findings**

The manuscript deals mostly with the natural history of a critically endangered and recently thought to be extinct of harlequin toad, which is an important contribution in an era where many species in this genus seem to be coming back. It certainly gives the

backbone for further studies, which could untanble their incredible comeback and coexistence with an emerging disease that caused their mass extinction in the first place. The conclusions are well stated, I would appreciate more on the conservation efforts, that given the authors experience they would recommend as stated above. Is this species now on solid footing? Should we try to be expanding its range artificially or is natural dispersal going to be enough for it to recover?

R: Thanks for the positive comments! We still do not have enough information to say if the species is stable, increasing or declining. We have expanded the Recommendations section to incorporate the suggested actions. See line numbers 488–543.

**Reviewer 2 (Stefan Lötters)**

**Comments**

Title: Here you refer to "harlequin toad", while elsewhere in the text you refer to "harlequin frog" – never "harlequin toad" but you use "toad". Chose one, both is possible but not within the same piece of text I would say.

R: Thanks for noticing this inconsistency. Now we use "toad" in the MS.

Line 58: I suggest removing key words that appear in the title.

R: The word *Atelopus* in the title has been removed.

Line 67: Say "2,800 to 4,200 m a.s.l."

R: Changes have been made as suggested.

Line 69: Provide full name of Jimenez de la Espada, which reads better, I think; otherwise place year and page after mentioning the name here and delete at the end of the sentence.

R: We included the complete name "Marcos Jimenez de la Espada".

Line 71: What does he describe with regard to reproductive habits?

R: He described that Jambatos perform an axillary amplexus and that the palms of their feet and hands become thicker and more extended, possibly preparing for egg laying in the water (Jiménez de la Espada, 1875).

Line 72: Rewrite: "Years later, in 1981, at the same site…".

R: Changes have been made as suggested.

Line 86: Move year in parentheses behind name and delete at the end of the sentence.

R: Changes have been made as suggested.

Line 87: Say "public monument", if true.

R: Changes have been made as suggested.

Line 89: "last record" not correct, restate like "last record for almost three decades" or so.

R: Changes have been made as suggested. See line numbers 90–91.

Line 94: Although La Marca (2005) is the first genus-wide study, the update of that study (Lötters et al. 2003) refer to more species (131 instead of 113) – including those undescribed or not identified in 2005 – and provides data on declines. Maybe cite both here.

R: Changes have been made as suggested.

Line 97: Say "amphibian skin fungus".

R: Changes have been made as suggested.

Line 103: Say "pathogenic fungus".

R: Changes have been made as suggested.

Line 122: I recommend to exclusively or in addition mention the full Spanish name of the ministry.

R: Changes have been made as suggested.

Line 126: Place year in parentheses.

R: Changes have been made as suggested.

Line 128: Say "2,812 m a.s.l."

R: Changes have been made as suggested.

Line 135: Say "4,064 m. a.s.l."

R: Changes have been made as suggested.

Line 138: You refer to WorldClim2 data here that are based on interpolations. You should refer to this circumstance, as interpolations in the Andes may easily have errors. Also mention the period for with these data are valid. See below my comment on line 259.

R: We agree with the reviewer. We have incorporated the following sentence (line number 139–141):

A limitation of the WorldClim climatic data is that they are based on interpolations and, given the topography of the Andes, some of these interpolations might have errors.

Line 139: You have super nice and informative illustrations including ventral aspects of the jambato. It would be informative to the reader to see what the species looks like from above (I mean the adult, not only the tadpole that you illustrate). Maybe add a specimen on white background to figure 1 here. Alternatively, maybe right in the beginning show a separate illustration of the frog in its natural habitat (a new figure 1), which provides a better impression of the study organism to the reader who is not so much into Atelopus.

R: In Figure 1 we have incorporated a photo of an individual of *Atelopus ignescens* in dorsal view.

Line 140f.: I think that he term "opportunistic" should be added somewhere in this paragraph, as this is a common term for how data were obtained.

R: Changes have been made as suggested.

Line 147: How many nocturnal surveys, at which hours? Rephrase "with the goal to find sleeping individuals". What about nocturnal activity of tadpoles? Did you look for tads at night?

R: Changes have been made as suggested.

Line 162, line 226: Which type of gloves?

R: We used different nitrile gloves for each individual (line number 165).

Line 167: Instead of "categories" better refer to the common term "landuse class" throughout the paper.

R: Changes have been made as suggested.

Line 168f.: What is a "spatial spread of areas"? You mean the spread of specimens within the area?

R: We have clarified the text.

Line 170: Not "1 km2" but "1 km²". BTW; this is quite coarse, given the general size of the area. Couldn't you downscale?

R: We have now generated a heat map of the records to convey the density of records of the species during the monitored year. (See Figure 5)

Line 178: What do you mean by "two lacking any presence record"? Please rewrite in way that you have not recorded the here. Maybe state how far the nearest ever captured specimen was to provide a better picture of apparent absence.

We have reworded the sentence.

Line 200: Amanda analyzed the photographs by visual inspection only? No software was used? You could easily provide them all in an electronic appendix here.

R: The photos were visually inspected (no software was used). We provide a link to an iNaturalist project with all the photos of the Jambatos.

Line 205: Say "2,971 m a.s.l."

R: Changes have been made as suggested.

Line 242: Add "a.s.l." again to the elevations mentioned here.

R: Changes have been made as suggested.

Line 259: Climate is defined as the mean weather conditions of the atmosphere over 30 years at a given site. I doubt the 51-year-period (1960-2021) given here for WorldClim 2.1. According to https://www.worldclim.com/version2, this period is 1970-2000.

R: For the climate analysis we have worked with monthly values of temperature and precipitation from 1960 to 2021. These data are available at: https://www.worldclim.org/data/monthlywth.html.

Line 265: I am not an expert and maybe I am wrong here. The longer the periods are, the more unnormalities are smoothened. Are the results the same when comparing e.g. only 1971-1980, 1981-1990, 1991-2000?

R: The main analyzes were made with the full temporal timeframe. The temporal partitions were made to show major population changes of the Jambato; for each time period we only estimated the average and annual standard deviation of the temperature and precipitation variables. Also, we included graphs (boxplots; Figure 12) to facilitate the visualization of climate in the three time periods.

Line 273: Instead of "recognized as different" say "considered to be likely different". BTW; I find this tricky, as, given figure 2, all 6 larvae were found over a distance of 1,000 m in the same stream and tadpole symbols in the figure even overlay suggesting they were close to each other. Honestly, we don't know much about larval movement or drift in Atelopus.

R: Changes have been made as suggested. We agree with the reviewer in that our knowledge on how tadpoles move or which microhabitats they prefer is very limited. Because of that, we do not discuss these aspects in the paper. However, for management proposes, it is important to mention that the Guambaine river is the only place, so far, where we have found tadpoles.

Line 276: Maybe add to "habitat" the term "landuse classes" in parentheses (see comment to line 167). This also applies to the text of figure 5 and to line 407f (and perhaps elsewhere in the text, figure legends, tables, appendices).

R: Changes have been made as suggested.

Line 323: As you also did the photo ID method to learn more about recaptures (cf. title of paragraph!) you state "our study was not designed with the aim of recapturing individuals". This is a bit unlucky, although I understand what you mean. However, this should be rewritten. I would especially recommend to emphasize that during the entire period you ONLY had 2 recaptures, which is remarkable.

R: Changes have been made as suggested. See line number 312–313. As we mention in the manuscript, the goal of this first phase was to understand the presence areas of the Jambato. Now, in collaboration with the Angamarca community, we are monitoring specific transects.

Line 328f: You mentioned above (MM) where photographs are deposited. Do not repeat here.

R: Changes have been made as suggested.

Line 353: Is there seasonal variation? Worth checking and mentioning. Maybe plot time against number of positives/negatives and show. Tadpoles were not tested for Bd, correct? Maybe mention in MM if not there.

R: We found 23 *Bd*-positive Jambatos belong to different months (December, April, May, and June). We have explained this better in the MS (see line number 322–327) and we have also created a new figure (Fig. 8) to visualize the number of individuals (males and females) infected by *Bd* in different months.

Line 354: What kind of signs of sickness would you expect? What did you look for? Maybe also mention in MM.

R: Changes have been made as suggested. See line numbers 325–327.

Line 380: Maybe consider to show Table S6 in the main paper, not the appendix. I find it interesting.

R: Changes have been made as suggested.

Line 383f.: This is what I mentioned before. There should be an effect of length of period, too!

R: As mentioned above, the main analyzes were made with the full temporal timeframe. The temporal partitions were made to show major population changes of the Jambato; for each time period we only estimated the average and annual standard deviation of the temperature and precipitation variables. Also, we included graphs (boxplots; Figure 12) to facilitate the visualization of climate in the three time periods.

Line 430: use the abbreviation "*Bd*" here.

R: Changes have been made as suggested.

Line 432: Is this prevalence among all *A. ignescens* samples over time? What about seasonal variation? What about prevalence of the entire amphibian community? Be clear here.

R: We have incorporated text (line numbers 397–417) to address these suggestions.

Line 435: "Lötters" not "Lotters" please, it makes a difference in German language.

R: Changes have been made as suggested.

Line 454: Maybe. But one may also see a direct link between the 1983 warm period and declines some years later. First, there is no robust data. What is 'the end of the 1980s'? Maybe it only then became more obvious to the people, as probably the species silently started to decline before. Second, what about delayed effects, e.g. a short life span combined with lack of reproductive success?

R: We address the Reviewer's suggestion in the following text (line numbers 426–436):

"Warm temperature by itself is insufficient to explain declines; according to Pounds et al. (2006) about 80% of the species that have disappeared were last sighted after relatively warm years and combined data point to the decline of the most vulnerable species towards the end of the 1980s. For example, our analyses show that the hottest year on record was 1983, but population crashes were not observed in Angamarca until the late 1980s, according to locals; however, it is possible that the species may have slowly declined as a result of higher adult mortality and low reproductive success. Likewise, when analyzing precipitation patterns during the pre-decline and population crashes periods (1960–1990), the driest year was 1979, again several years before observed declines. Thus, given these data, a most likely scenario explaining the Jambato declines is associated with the arrival and dispersal of Bd, as observed in other species across the globe (Scheele et al., 2019, 2020)."

Line 692: "Köhler", not "Köehler" please.

R: Changes have been made as suggested.
* * *
Again, we sincerely thanks the reviewers for the positive comments and recommendations. We have made an effort to address all the observations and hope that the Editor finds our answers well-sustained.

We look forward to the next steps of the process.

Sincerely,

Mateo and Juan

(On behalf of all authors)